# Fast growth of large-grain and continuous MoS$_2$ films through a self-capping vapor-liquid-solid method

Ming-Chiang Chang [1,2,9], Po-Hsun Ho [3,4,9✉], Mao-Feng Tseng [1,2,9], Fang-Yuan Lin[1,5,9], Cheng-Hung Hou [6], I-Kuan Lin[7], Hsin Wang[1,7], Pin-Pin Huang[1,5], Chun-Hao Chiang[7], Yueh-Chiang Yang [2], I-Ta Wang[7], He-Yun Du [8], Cheng-Yen Wen[3,7], Jing-Jong Shyue [6], Chun-Wei Chen[3,7], Kuei-Hsien Chen [1✉], Po-Wen Chiu [1,2✉] & Li-Chyong Chen [3,8✉]

Most chemical vapor deposition methods for transition metal dichalcogenides use an extremely small amount of precursor to render large single-crystal flakes, which usually causes low coverage of the materials on the substrate. In this study, a self-capping vapor-liquid-solid reaction is proposed to fabricate large-grain, continuous MoS$_2$ films. An intermediate liquid phase-Na$_2$Mo$_2$O$_7$ is formed through a eutectic reaction of MoO$_3$ and NaF, followed by being sulfurized into MoS$_2$. The as-formed MoS$_2$ seeds function as a capping layer that reduces the nucleation density and promotes lateral growth. By tuning the driving force of the reaction, large mono/bilayer (1.1 mm/200 μm) flakes or full-coverage films (with a record-high average grain size of 450 μm) can be grown on centimeter-scale substrates. The field-effect transistors fabricated from the full-coverage films show high mobility (33 and 49 cm$^2$ V$^{-1}$ s$^{-1}$ for the mono and bilayer regions) and on/off ratio (1 ~ 5 × 10$^8$) across a 1.5 cm × 1.5 cm region.

[1] Institute of Atomic and Molecular Sciences, Academia Sinica, Taipei 10617, Taiwan. [2] Department of Electrical Engineering, National Tsing Hua University, Hsinchu 30013, Taiwan. [3] Center of Atomic Initiative for New Materials, National Taiwan University, Taipei 106, Taiwan. [4] Department of Electrical Engineering, Stanford University, Stanford, CA 94305, USA. [5] Department of Chemistry, National Taiwan Normal University, Taipei 116, Taiwan. [6] Research Center for Applied Sciences, Academia Sinica, Taipei 11529, Taiwan. [7] Department of Materials Science and Engineering, National Taiwan University, Taipei 106, Taiwan. [8] Center for Condensed Matter Sciences, National Taiwan University, Taipei 10617, Taiwan. [9] These authors contributed equally: Ming-Chiang Chang, Po-Hsun Ho, Mao-Feng Tseng, Fang-Yuan Lin. ✉email: river770323@gmail.com; chenkh@pub.iams.sinica.edu.tw; pwchiu@ee.nthu.edu.tw; chenlc@ntu.edu.tw

Apart from graphene, transition metal dichalcogenides (TMDs) with atomic thickness are the most renowned two-dimensional (2D) materials because of their excellent electrical and optical properties[1–6]. Their robust physical properties in atmosphere enable their practical applications in novel optoelectronic devices[7,8]. For electronics, TMDs with atomic thickness, which inherently have no surface dangling bonds, are immune to mobility degradation and short channel effects in contrast to conventional three-dimensional materials, such as Si and GaAs[9–11]. Such materials have layer-dependent bandgaps from near infrared to visible regions[12,13], and thus, TMDs are favorable for energy or optical applications[7,8,14,15]. Furthermore, the difference between Berry curvature at the K and K′ valleys of monolayer TMDs generates new opportunities for valleytronics[16,17]. Despite these remarkable properties, TMDs still have limitations arising from spatial nonuniformity. Therefore, the fabrication of high-quality and large-grain films is thus crucial for TMDs. Currently, chemical vapor deposition (CVD) is the most recognized method for producing high-quality monolayer TMDs because of its low cost and scalability[18–22]. Conventional CVD methods of TMD fabrication are based on the reaction of gas-phase chalcogens (e.g., S and Se) and metal oxides (e.g., $MoO_3$ and $WO_3$)[18–22]. Generally, in gas-phase reactions, the grain size of TMDs is limited by the high nucleation density and typically is <500 μm. Recently, researchers have used various methods, which include fabricating at a high temperature[23], inserting diffusion barriers[24], and using an extremely small amount of precursors[25], to reduce nucleation density and to increase the surface diffusion length for growing large TMD crystals. These methods can produce comparatively large crystals but considerably reduce the coverage of TMD crystals[25], which hinders their practical applications.

For bulk materials, Czochralski[26,27] method enables the production of a single-crystal ingot with a diameter of up to 300 mm by vertically pulling a solid seed crystal from a liquid source[28]. This method provides the unprecedentedly high uniformity of conventional bulk materials at an ultra-large scale. Moreover, a liquid source can more easily dissolve other solid dopants than a gas–gas reaction[29]. Therefore, it is desirable to grow solid crystals from liquid sources. For the growth of TMDs, Li et al. recently proposes the vapor–liquid–solid (VLS) reaction for fabricating high-quality $MoS_2$ nanoribbons from a liquid precursor on a sodium chloride (NaCl) single crystal[30]. First, NaCl reacts with $MoO_3$ to form a eutectic compound ($Na_2Mo_2O_7$), which has a relatively lower melting point and exists in a liquid phase under growth conditions (generally 700–800 °C). Second, the sulfur vapor is rapidly dissolved into the liquid and reacts to form a solid-state monolayer $MoS_2$ on NaCl. However, because of the low wettability between the NaCl and liquid $Na_2Mo_2O_7$ droplets, this method can only generate $MoS_2$ nanoribbons, which considerably limits its application. This problem can be solved by growing on other substrates with a better wettability[31].

Herein, a self-capping vapor–liquid–solid (SCVLS) reaction, which can grow large single crystals and full-coverage TMD films, is proposed. A solid precursor comprising ultra-thin $MoO_3$, $SiO_2$, and NaF layers was used for the controllable eutectic reaction of $MoO_3$ and NaF at high temperature. The as-formed eutectic liquid ($Na_2Mo_2O_7$) rose to the surface and was sulfurized into $MoS_2$ seeds. These seeds, acted as a self-capping layer, redirected the rising liquid into a horizontal direction. The residual liquid was continuously pushed along the growth direction and eventually sulfurized to form new $MoS_2$ at the edge of the $MoS_2$ seeds. This growth mechanism enables fabrication of ultra-large (~1.1 mm) single crystals. Moreover, continuous large-area $MoS_2$ film with large-grain size (~450 μm) can also be fabricated using thicker precursor. By controlling the kinetic factors of this reaction, the layer number can be controlled and large bilayer $MoS_2$ (~200 μm) can be achieved. In this study, the quality and uniformity of $MoS_2$ grown using this method are evaluated through electron microscopy, optical spectroscopy, and electrical measurements. For electrical measurements, both mono- and bilayer $MoS_2$ field-effect transistors (FETs) show high mobility (33 and 49 $cm^2V^{-1}s^{-1}$), large on/off ratio ($5 \times 10^8$), and high current density (up to 230 and 390 μA μm$^{-1}$). The large-grain, continuous film exhibits high performance across a 1.5 ×1.5 cm area, making the SCVLS method promising for practical applications.

## Result

**Material synthesis and growth mechanism.** Figures 1a and S1 show that a smooth $MoO_3$ layer was grown on c-plane sapphire through plasma-enhanced atomic layer deposition (PEALD). $SiO_2$ and NaF layers were stacked on top of the $MoO_3$ layer through sputtering and thermal evaporation, respectively. The $SiO_2$ layer acted as a diffusion membrane to control the amount of $MoO_3$ vapor that broke the $SiO_2$ layer (Fig. 1b and Supplementary Fig. 2), diffused upward and reacted with the NaF layer at a temperature higher than 500 °C to form liquid-phase $Na_2Mo_2O_7$ and gas-phase $MoO_2F_2$ (Fig. 1b and Supplementary Fig. 3). Simultaneously, the consumption of NaF generated holes and pathways in the NaF layer, which allowed $Na_2Mo_2O_7$ and $MoO_2F_2$ to gradually rise to the top surface of the NaF through the pressure gradient and capillary phenomenon (Fig. 1c). Meanwhile, sulfur vapor was introduced into the system and rapidly dissolved in the eutectic liquid ($Na_2Mo_2O_7$) that rose to the surface. As discussed in the first VLS paper, the products of this VLS reaction were $MoS_{2(s)}$ and sulfur oxides ($SO_{2(g)}$ and $SO_{3(g)}$)[30]. Moreover, the molten liquid surface provided a temporarily atomic-flat and defection-free surface with a low nucleation density[23,32]. The oversaturated $MoS_2$ precipitated as seed layers on the liquid surface (Fig. 1e). The as-formed $MoS_2$ seed layers blocked the route for sulfur to dissolve into the liquid and redirected the underlying liquid to move horizontally. The unsaturated liquid then emerged to the surface at the $MoS_2$ edge, and this was where the SCVLS reaction primarily occurred. Therefore, $MoS_2$ laterally grew into large crystals (Fig. 1f, g). Millimeter-sized $MoS_2$ single crystals were obtained using this SCVLS method. The large triangular $MoS_2$ flakes are single crystals in nature, as validated by diffraction analysis at multiple spots in a large flake (Supplementary Fig. 4). The zoom-in image of a $MoS_2$ edge exhibits many bilayer fringes (Fig. 1h), which were a result of the precipitation of the residual liquid at the edges of the $MoS_2$ flakes during the rapid cooling process. These fringes validate the existence of the liquid phase during growth and the aforementioned mechanism. In contrast to the conventional CVD method, wherein the nonuniform gas flow in the furnace often gives resultant film of poorer uniformity[33], extending the SCVLS method to a wafer scale is facile because the precursor rises uniformly from the bottom surface of the growth substrate. Supplementary Fig. 5 shows a full-coverage $MoS_2$ film grown on a sapphire of $3 \times 3$ cm$^2$ (size was only limited by the CVD tube size). This process can also be extended to grow $MoSe_2$ by replacing sulfur with selenium, result of which is shown in Supplementary Fig. 6.

**Characterization of as-grown material.** Compared with the conventional CVD technique, the final products of this SCVLS method were more complex (Fig. 2a), which comprised two parts: the top $MoS_2$ layer and the complex solid products generated by the quenched liquid within the NaF matrix. During the sulfurization process, sulfur vapor was primarily dissolved into the liquid through the exposed liquid–gas

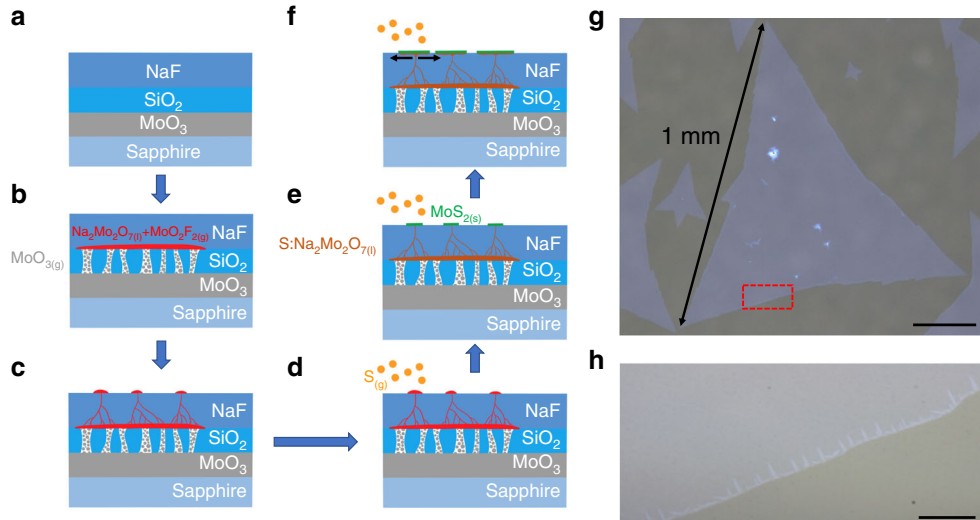

**Fig. 1 Schematics of SCVLS growth mechanism and the grown MoS₂. a** Structure of the solid precursor used for the SCVLS method. **b** At growth temperature, MoO₃ vaporized and penetrated through the SiO₂ diffusion membrane. MoO₃ and NaF reacted to form liquid-phase Na₂Mo₂O₇ (colored in red) at the growth temperature. **c** Through reactive digging and the capillary effect, the liquid precursor gradually rose to the NaF matrix surface. **d** Sulfur vapor was introduced into the system and started to dissolve into the Na₂Mo₂O₇ liquid. **e** Liquid precursor sulfurized into the MoS₂ seed layer. **f** Capped by the MoS₂, the emerging liquid was redirected horizontally and converted into MoS₂ when it contacted and dissolved sulfur vapor at the edge of the MoS₂ flakes. **g** A 1-mm MoS₂ flake grown through the SCVLS method. Scale bar is 200 μm. **h** Magnified image of the MoS₂ grain edge. The fringes at the edge indicate the presence of the liquid precursor during the growth process. Scale bar is 20 μm.

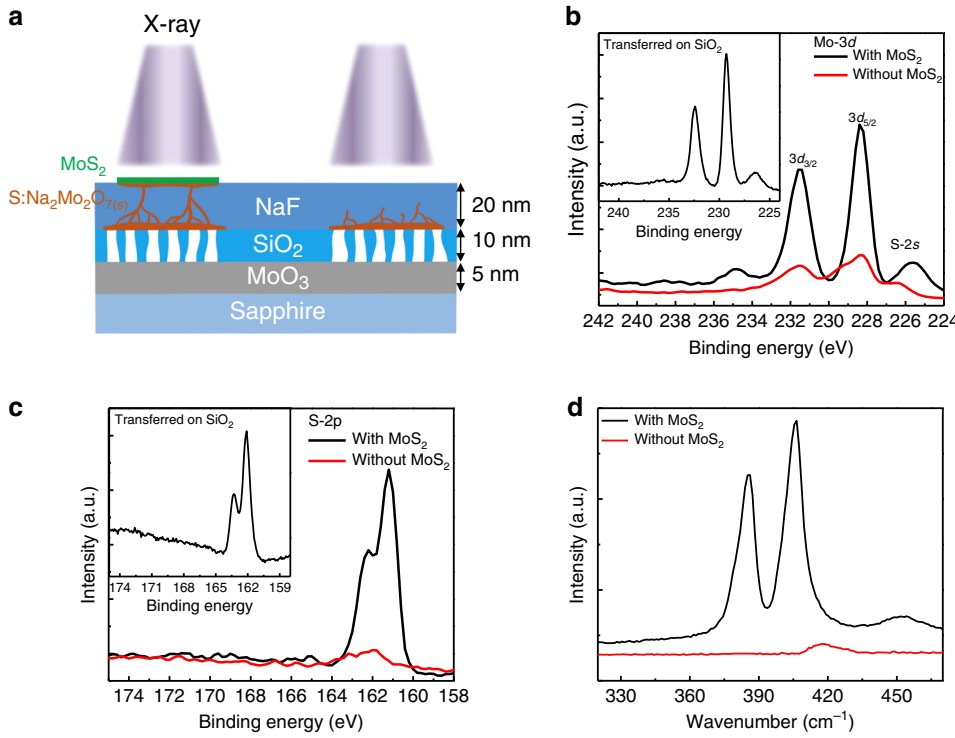

**Fig. 2 Characterization of the MoS₂ monolayer and remaining solid precursors. a** Schematic of the sampling areas of XPS. XPS data of **b** Mo-3$d$ and **c** S-2$p$ on sites that were covered (black) and not covered (red) with MoS₂. The broad peak of Mo-3$d$ at sites that were not covered with MoS₂ indicates the complex chemical environment of Mo below the surface. Insets in **b** and **c** are the Mo-3$d$ and S-2$p$ of the MoS₂ film transferred onto a SiO₂ substrate. **d** Raman spectra taken form positions with (black) and without (red) MoS₂ coverage.

interface, and thus, the oversaturated liquid could continuously precipitate MoS₂ at the edge of the MoS₂ seeds. However, the reactions were not limited to the top surface. With a lower sulfur concentration, some incomplete sulfurization reactions

and precipitations were observed within the NaF matrix (Supplementary Fig. 7). Upon cooling, the unsaturated liquid solidified and resided below the MoS₂ flakes or was buried in the NaF matrix (Fig. 2a). X-ray photoelectron spectroscopy (XPS)

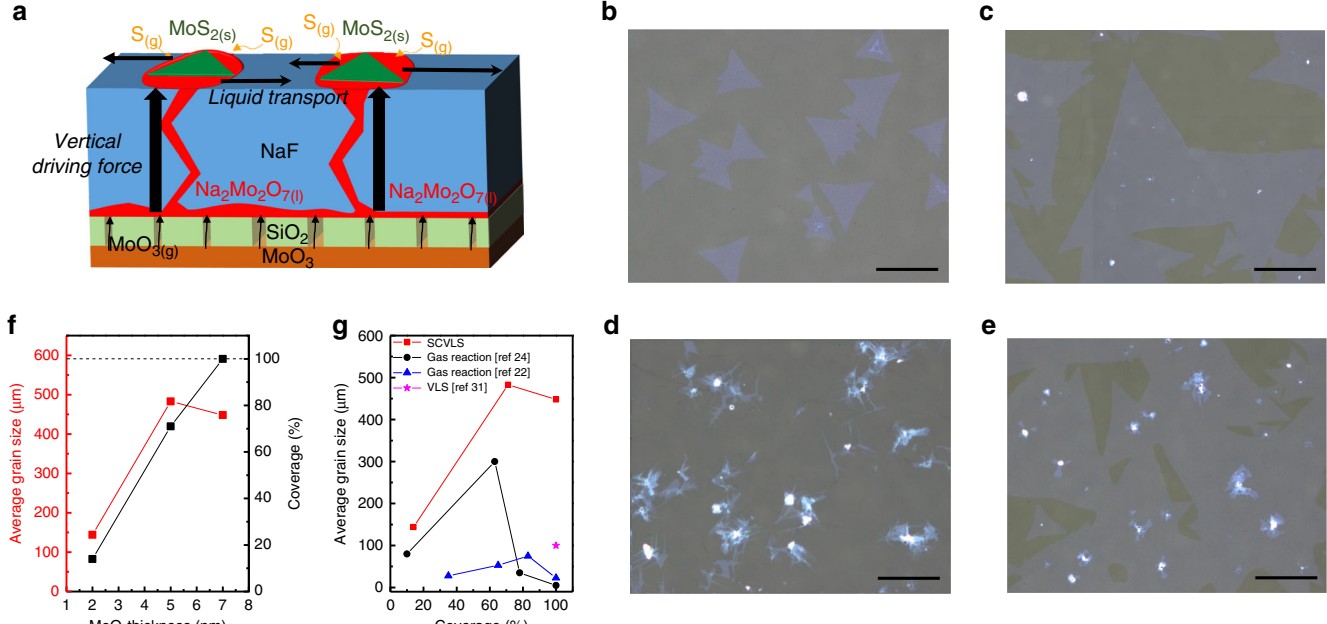

**Fig. 3 Driving forces of SCVLS reaction. a** Schematic of the vertical driving force and horizontal liquid transport. Optical microscopy images of $MoS_2$ grown using **b** 2-nm **c** 5-nm, and **d** 7-nm $MoO_3$ as a solid precursor. The growth time was 10 min. **e** 1-min growth using the same precursor sample as in **d**. The scale bars in **b**, **d**, and **e** are 200 μm and the one in **c** is 300 μm. **f** Average grain size and coverage of the $MoS_2$ flakes as a function of $MoO_3$ precursor thickness. **g** Comparison of SCVLS, VLS, and gas-phase CVD. SCVLS reaction enables a relatively large-grain size when a full-coverage film was achieved.

was used to analyze the final products. Figure 2b, c shows Mo-$3d$ and S-$2p$ spectra obtained from regions covered by large $MoS_2$ flakes and regions with the exposed NaF matrix, respectively. For a region covered with $MoS_2$, the Mo-$3d$ peak of ~230–228 eV could be deconvoluted into sharp $Mo^{4+}$ and broad $Mo^{x+}$. The $Mo^{4+}$ signal was obtained from the top $MoS_2$, and the $Mo^{x+}$ signal was obtained from the precipitates and solidified liquid phase in the NaF matrix, as shown using an XPS depth profile (Supplementary Fig. 7). With a lower sulfur concentration in the matrix, the possible products for the precipitate and quenched liquid were amorphous $MoS_2$, $MoO_2$, and $MoS_xO_y$, comprising the $Mo^{x+}$ signal. In addition, a small peak at ~235 eV indicates $Mo^{6+}$, which is the peak from $Na_2Mo_2O_7$, indicating the presence of residual unreacted precursor below $MoS_2$; this supports the as-proposed horizontal transport of the liquid. For a region without $MoS_2$, only a small amount of sulfur diffused into the NaF matrix and reacted with the liquid below the surface. Figure 2b, c shows a considerably weaker $Mo^{x+}$ and sulfur signal, which indicates that no liquid rose to the top surface. The resultant products were further characterized using Raman spectroscopy (Fig. 2d). The region covered with $MoS_2$ exhibited sharp $E_{2g}$ and $A_{1g}$ peaks with a spacing of 19.5 cm$^{-1}$, thus validating the high quality and monolayer characteristics for the as-grown $MoS_2$. The region that was not covered by $MoS_2$ exhibited no significant Raman signal, which validated the absence of any crystalline product in the NaF matrix. The as-grown monolayer $MoS_2$ could be readily transferred to various substrates using the conventional polymethyl-methacrylate (PMMA) method. Atomic force microscopy images and photoluminescence spectrum in Supplementary Fig. 8 and 9 also confirm the monolayer property. The insets of Fig. 2b, c show the XPS results of the as-transferred $MoS_2$ on silica. Compared with the as-grown sample, the transferred $MoS_2$ exhibited sharp and clean $Mo^{4+}$ signals at 229.3 and 232.4 eV. The Mo–S ratio, which was calculated by integrating the area of the Mo and S signals, was 1:2, as expected for high-quality $MoS_2$.

**Controlling the coverage and thickness of $MoS_2$.** With a suitable sulfur source, the growth rate of SCVLS process was controlled by horizontal transport rate of the liquid. Rapid horizontal mass transport of the liquid was crucial for growing large monolayer $MoS_2$. The driving force stemmed from the diffusion and capillary phenomena of the high-pressure liquid and gas produced by the eutectic reaction (Fig. 3a). Because $MoS_2$ layers covered the top surface and confined the liquid flow, the vertical driving force was redirected horizontally, and both vertical and horizontal liquid transport was promoted with increasing amount of liquid source. The rapid SCVLS reaction could thus be performed under this condition, and the rapid growth rate abruptly increased the grain size of $MoS_2$. However, when the driving force was weak, the low growth rate would result in more nucleation seeds on NaF, thus reducing the average size of the $MoS_2$ crystals. In some regions, the weak driving force was not sufficient to push the liquid to the surface. The sulfur vapor would slowly diffuse into the NaF matrix, react with the liquid, and eventually solidify. Therefore, no $MoS_2$ was grown on the surface under this condition, and this phenomenon reduced the coverage of $MoS_2$. Here, the vertical driving force was controlled using different amounts of $MoO_3$ sources. Figure 3b–d shows the optical images of the as-grown $MoS_2$ with different thicknesses of $MoO_3$ precursor layers. The grain size and coverage of $MoS_2$ abruptly increased with the increasing thickness of $MoO_3$ precursor (Fig. 3f). In order to estimate the grain size of the full-coverage film (Fig. 3d), the growth time was reduced from 10 (Fig. 3d) to 1 min (Fig. 3e), to monitor the grain size before grains merged into continuous film. The average grain size of the full-coverage film was ~450 μm, which is the largest recorded average grain size for completely covered $MoS_2$ film. Although there are some thick islands on the film (Fig. 3d and Supplementary Fig. 10), this large-grain continuous film can still demonstrate outstanding electrical performance shown in the later section. The trend of the coverage and average grain size versus the precursor thickness in SCVLS (Fig. 3f) is very different from that for the common gas-phase reaction CVD. For gas-phase CVD, the average grain size

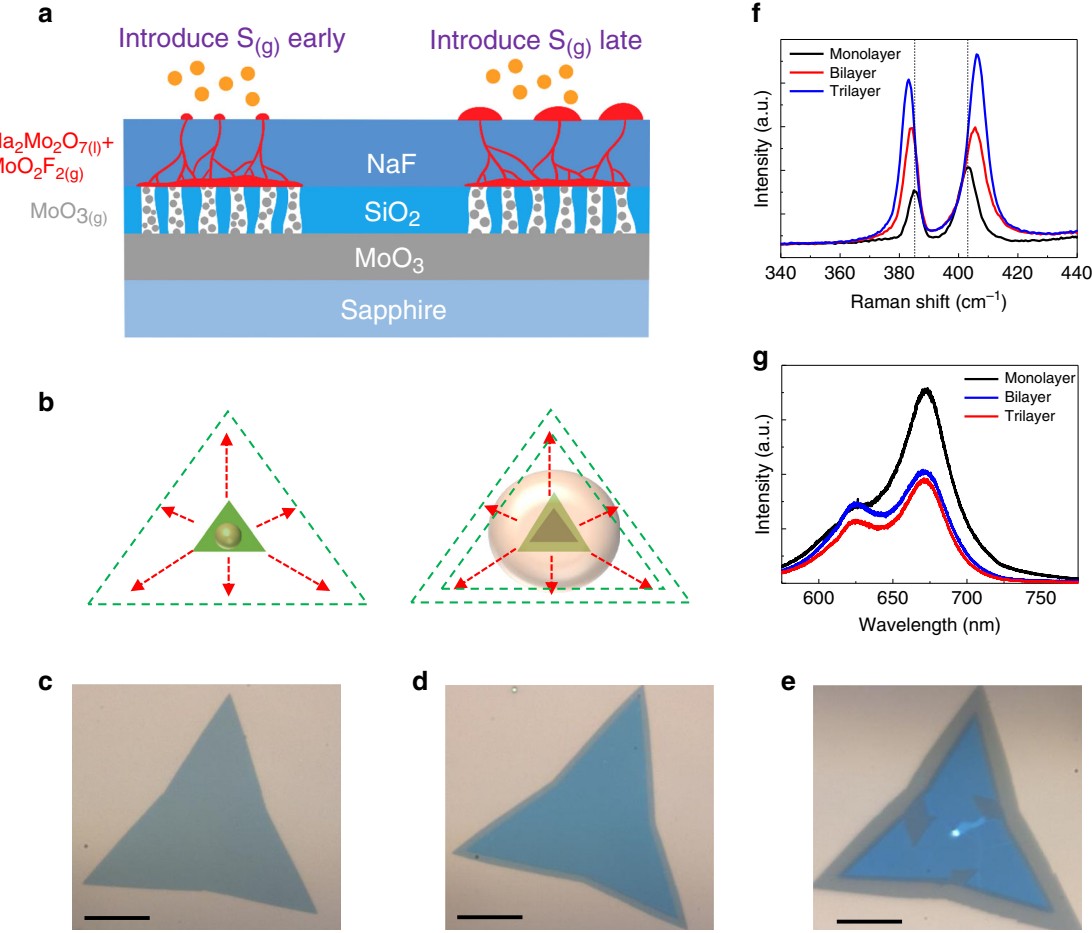

**Fig. 4 Dynamic effect on SCVLS method. a** Schematic of the timing of introducing sulfur vapor affecting the final growth product. **b** When sulfur vapor was introduced early, the $Na_2Mo_2O_7$ precursor rapidly formed the $MoS_2$ seed layer when exposed on the surface (left). When sulfur vapor was introduced later, the $MoS_2$ seed layer was formed at the solid–liquid interface, thus leaving a droplet of the liquid precursor on top of the interface. This droplet was later sulfurized into the second layer of $MoS_2$ (right). Optical images of the transferred **c** monolayer, **d** bilayer, and **e** trilayer $MoS_2$. Scale bars are 50 μm. **f** Raman and **g** photoluminescence spectra of the monolayer, bilayer, and trilayer $MoS_2$.

of the continuous film is reduced by a factor of 10–100 compared with the largest isolated crystals because the larger amount of the precursor for growing continuous film abruptly led to the increased nucleation density and thus reduced grain size (Fig. 3g)[25,34]. However, for SCVLS method, the average grain sizes of the continuous film (450 μm) and largest isolated crystals (500 μm) are similar because of the self-capping effect and the fast transport of liquid. Furthermore, coverage of ~82% was reached within 1 min of fabrication (Fig. 3e), which demonstrates the rapid growth rate (370 μm/min, see Supplementary Fig. 11). This may be a result from the fast transport of liquid assisted by the fluoride surface, which has been shown to enhance growth rate of 2D materials[35]. The driving force and nucleation density could be further controlled by tuning the thickness of $SiO_2$ membranes, growth temperature, and the thickness of NaF (Supplementary Figs. 12–14).

In addition to large-grain continuous film, layer-controlled growth of bilayer and multilayer are also attractive because of the better electrical performance of few-layer $MoS_2$[36–39]. SCVLS can also control the layer number of $MoS_2$ with its special growth mechanism we proposed. In the growth condition of previous paragraphs, sulfur vapor was introduced before or during the liquid's rise to the surface. The small liquid droplets rapidly dissolved sulfur and formed monolayer $MoS_2$ capping seeds, which promoted the horizontal mass transport and formed large

monolayer $MoS_2$ grains. It is noteworthy to mention that the SCVLS reaction can be dynamically controlled by changing the timing of sulfurization (Fig. 4a). When sulfur vapor was introduced later, the emerged liquid would form into a large droplet (Supplementary Fig. 15). During sulfurization, the as-formed small $MoS_2$ seeds were buried in the oversaturated liquid. Under this condition, fresh $MoS_2$ could be formed at the edge of the original seeds, and a second layer could be grown on the $MoS_2$ seeds (Fig. 4b). Large bilayer $MoS_2$ crystals could be fabricated by delaying the sulfurization timing for 2 min. Trilayer $MoS_2$ was occasionally observed when sulfurization is delayed. Figure 4c–e are the optical images of the transferred mono-, bi-, and trilayer $MoS_2$ on $SiO_2$/Si substrates, respectively. The clear optical contrast shows the characteristics of the mono-, bi-, and trilayers of each $MoS_2$. AFM images in Supplementary Fig. 16 also confirm the thickness of these samples. Figure 4d, e shows that the second and third layers are well-aligned with the bottom $MoS_2$ layer, thus indicating the epitaxial growth of an excess $MoS_2$ layer. The diffraction patterns manifest a 2H-type stacking order of the SCVLS reaction (Supplementary Fig. 17)[30]. Raman spectra in Fig. 4f further validate the layer number and strong interaction between the mono-, bi-, and trilayer $MoS_2$ grown using the SCVLS method. The peak separations of $E_{2g}$ and $A_{1g}$ are 18.0, 21.5, and 23.2 $cm^{-1}$, which are similar to the values previously reported for exfoliated $MoS_2$[40]. The strong interaction between

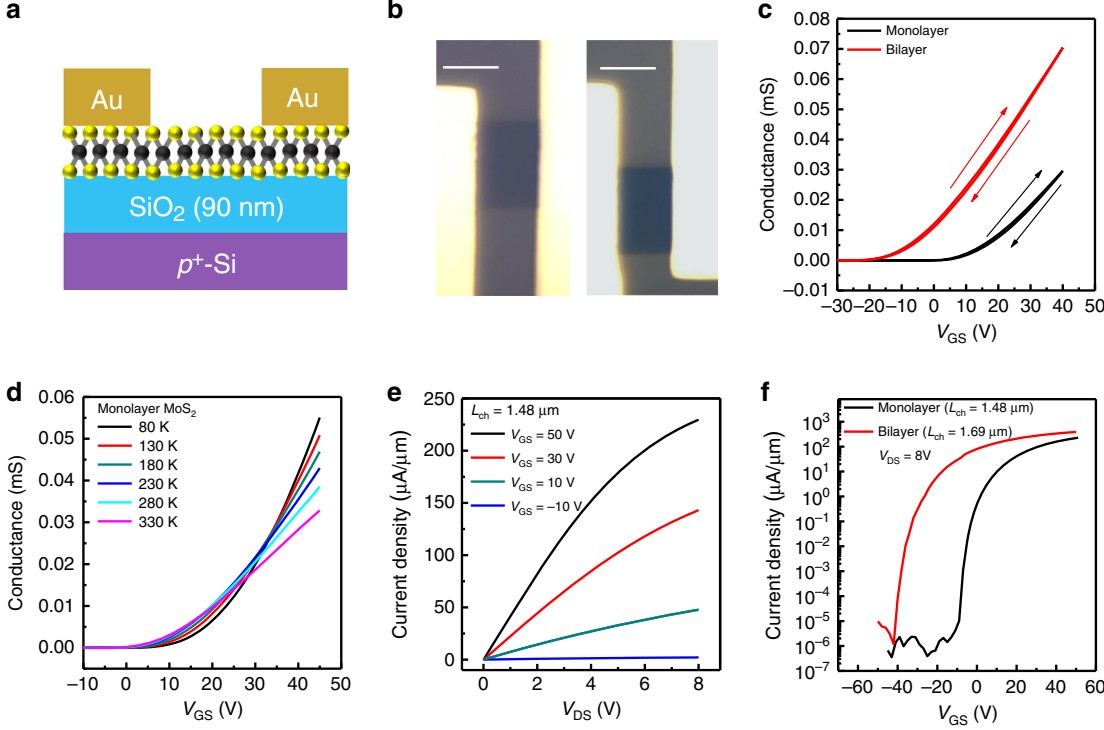

**Fig. 5 Transport properties of the MoS₂ grown through the SCVLS method. a** Schematic image of a back-gate monolayer MoS₂ transistor. **b** Optical image of a monolayer and bilayer FETs. Both scale bars are 5 μm. **c** Gate-dependent conductance of devices shown in **b**. **d** Temperature-dependent transport property of the monolayer MoS₂ FET shown in **b**. A clear MIT is observed at $V_{GS}$ of 30 V. **e** $V_{DS}$-dependent source-drain current density of the monolayer device at different gate voltages. The channel length is 1.48 μm. **f** Log plot of the gate-dependent current density of the short channel mono- and bilayer devices under an 8-V source-drain bias. The on/off ratio is $5 \times 10^8$.

each layer changes the dielectric environment, thus softening the in-plane $E_{2g}$ mode (red-shift). Moreover, the strong interaction between interlayer S increases the restoring force, thus stiffening the out-of-plane $A_{1g}$ mode (blue shift)[40]. The photoluminescence spectra in Fig. 4g exhibit clear quenched signals for bilayer and trilayer MoS₂ because of the direct–indirect band gap transition for monolayer and bilayer MoS₂[4,6]. By employing dynamic control of sulfurization, a large bilayer single crystal (200 μm) is successfully synthesized, the grain size of which is comparable to the large bilayer crystals in the previous studies[37,38]. Moreover, the ability to change the layer number of MoS₂ by controlling the size of the droplet validated the SCVLS mechanism proposed in Fig. 1.

**High-performance FET device.** The electrical properties of the large monolayer MoS₂ crystal grown by the SCVLS method was examined by measuring the transport properties of MoS₂ FETs. Figure 5a shows an FET with a back-gate structure and 90-nm-thick SiO₂. Both mono- and bilayer MoS₂ FETs were fabricated as shown in Fig. 5b and Supplementary Fig. 18. Figure 5c displays the gate-dependent conductance of the MoS₂ FETs. Both devices have a very small hysteresis, indicating low defect and impurity induced trap density in the SCVLS growth and the device fabrication processes. The field-effect mobility was calculated using $\mu_{FE} = \frac{L_{CH}}{W}\frac{1}{C_G}\frac{dG}{dV_{GS}}$, where $C_G$, $L_{CH}$, $W$, $V_{GS}$, and $G$ stand for the back-gate capacitance, channel length, channel width, back-gate voltage, and sheet conductance of the channel, respectively. Because of its higher carrier density and stronger charge screening effect, bilayer MoS₂ has a smaller threshold voltage ($V_{th}$) and higher mobility. The mobilities of the mono- and bilayer MoS₂ FETs are 33 and 49 cm² V⁻¹ S⁻¹, respectively.

These values are comparable to the exfoliated MoS₂[41] and the best reported values of CVD MoS₂ on SiO₂[38,42,43], showing the high quality of MoS₂ grown through the SCVLS method. The temperature-dependent transport also confirms the quality of SCVLS MoS₂ as shown in Fig. 5d. For the monolayer device, a clear metal–insulator transition (MIT) was observed, which is generally detected when using a high-$k$ dielectric layer to reduce Columbic scattering in the MoS₂ channel[2]. For back-gate devices without a high-$k$ dielectric layer, a clear MIT occurs only when using high-quality MoS₂ with a low concentration of sulfur vacancies. In general, according to the Ioffe-Regel criterion[44], MIT occurs when the critical channel conductance is approximately one quantum conductivity ($e^2/h$)[2,44]. If the crossover point is in a lower carrier concentration region, this directly reflects the high-mobility property of MoS₂[45,46]. The carrier concentration of the transition point is calculated using

$$n_{MIT} = \frac{C_G}{e}(V_T - V_{MIT}), \quad (1)$$

where $V_T$ and $V_{MIT}$ are the threshold voltage and voltage at which the MIT occurs, respectively[47,48]. The $n_{MIT}$ of the SCVLS MoS₂ in this study is $4.3 \times 10^{12}$ cm⁻², which is even lower than the previously reported intrinsic exfoliated MoS₂ with low S-vacancy concentration[47]. This indicated the high quality and low sulfur vacancy concentration of the MoS₂ fabricated using the SCVLS method. Figure 5e is the output characteristics of a short channel (1.48 μm, see Supplementary Fig. 19) monolayer FET at various back-gate voltages. The linear behavior in the low source-drain voltage ($V_{DS}$) region shows a good Ohmic property of contacts. Figure 5f presents the semi-logarithmic of the gate-controlled current density. The device on/off ratio can reach $5 \times 10^8$, with a subthreshold swing of 980 mV dec⁻¹. The maximum current

**Table 1 Comparison of the different CVD methods.**

| Growth method | Maximum grain size (single crystal) | Mobility (cm² V⁻¹ S⁻¹) | Maximum current density (μA μm⁻¹) | On/off ratio | Reference |
|---|---|---|---|---|---|
| CVD MoS₂ (MoO₃ + S, substrate control)[49] | 200 μm | 25 | | $10^7$ | ACS Nano 2015, 9, 4611 |
| CVD MoS₂ (MoO₃ + S, flow control)[50] | 300 μm | 30 | | $10^6$ | Adv. Sci. 2016, 3, 1500033 |
| CVD MoS₂ (MoCl₅ + DMS, NaCl catalyst)[51] | 50 μm | 10.4 | | $10^7$ | Nanotechnology 2017, 28, 465103 |
| CVD MoS₂ (MoO₃ + S, molten Na:glass)[32] | 563 μm | 24 | 123 | $10^9$ | Appl. Phys. Lett. 2018, 113, 202103 |
| CVD MoS₂ (MoO₃ + S with PTAS salt)[42] | 200 μm | 35 | 270 | $10^7$ | Nano Lett. 2018, 18, 4516 |
| CVD MoS₂ (Mo foil + S, soda–lime glass)[52] | 400 μm | 11.4 | | $10^6$ | Nat. Commun. 2018, 9, 979 |
| SCVLS MoS₂ | 1.1 mm | 33 | 230 | $5 \times 10^8$ | This work |

All transport data were obtained from the back-gate, monolayer MoS₂ FETs for comparison.

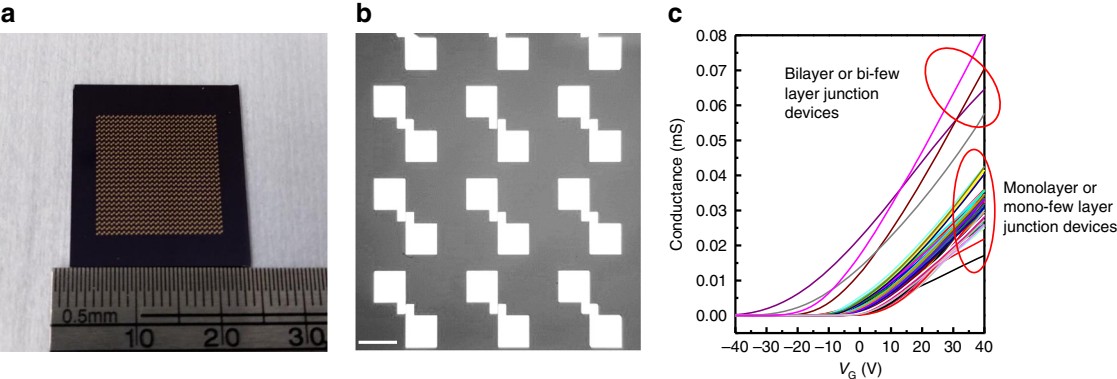

**Fig. 6 Transport properties of the large-grain, continuous film. a** Photo-image of the as-fabricated FETs across a 1.5 × 1.5-cm region. **b** Optical image of FET devices. The scale bar is 150 μm. The fine feature is shown in Supplementary Fig. 18. **c** Gate-dependent conductance of the devices across the large area.

density in the monolayer MoS₂ is 230 μA μm⁻¹ (390 μA μm⁻¹ for bilayer), potentially comparable to the optimal reported in consideration of the difference in contact geometry[38,42]. Table 1 lists the recently reported monolayer MoS₂ back-gate FETs. MoS₂ grown by the SCVLS method exhibits the largest grain size and remarkable electrical performance compared with other CVD techniques. The good uniformity of the large monolayer crystal is confirmed by measuring 18 FETs in a 1-mm crystal (Supplementary Fig. 20). In addition, with the capability of growing large-grain and continuous film, we fabricated hundreds of devices across a 1.5 × 1.5-cm area, as shown in Fig. 6a, b. Figure 6c is the gate-dependent conductance of a hundred devices in the whole area. Ninety percent of devices have pure monolayer channel and show high mobilities ($34 \pm 7$ cm² V⁻¹ s⁻¹) with small variation of $V_{th}$ ($4.9 \pm 2.3$ V). Devices have larger variation in mobility and $V_{th}$ if their channels are bilayer, few-layer, mono-few-layer junction, or monolayer with a small few-layer flake on top. However, the mobilities are still high for all of the devices because of the large-grain monolayer underneath (Supplementary Fig. 21). This demonstrates the advantage of using the SCVLS method for practical applications.

## Discussion

In summary, a new concept of growing high-quality single-crystal 2D materials from the liquid precursor was proposed using the SCVLS method. The rapid horizontal mass transport promotes the lateral growth of 2D materials and allows the growth of MoS₂ flakes as large as 1.1 mm. The self-capping effect drastically reduces the nucleation density even under large amount of precursor and results in a wafer scale, 100% coverage MoS₂ films with an average grain size of up to 450 μm. It overcomes the bottleneck of the conventional gas-phase CVD reaction, which is the trade-off between coverage and grain size. The quality and

uniformity of the as-grown MoS₂ were carefully evaluated through the electrical properties of MoS₂ FETs across a large area. High-mobility MoS₂ devices are demonstrated across a 1.5 × 1.5-cm area. Moreover, this method is capable of fabricating large bilayer MoS₂ crystals by controlling the timing of sulfurization. More sophisticated sulfurization precursor such as H₂S is expected to improve the layer number control or the uniformity of continuous films. Fabricating crystals by using the liquid–solid reaction, such as in doping and alloying, is one expected niche of this SCVLS method, which provides a new approach for synthesizing industrial-grade 2D materials for practical applications in 2D electronics.

## Methods

**Preparation of solid precursor.** 3 × 3-cm c-plane (0001) sapphire substrate was cleaned first by deionized water then sonicated in acetone and isopropyl alcohol for 20 and 5 min, respectively. MoO₃ film with well-controlled thickness was grown on top of sapphire substrates with a homemade PEALD system using Mo(CO)₆ as precursor and oxygen plasma as the oxidation reactant. For each deposition cycle, a Mo(CO)₆ precursor pulse is provided into the chamber, then the excess precursor is purged away by argon, and finally oxygen plasma (up to 200 W) is used to oxidize the precursor and form uniform MoO₃ film. Thermal evaporation can be used to replace the PEALD process for depositing MoO₃ but will result in worse MoS₂ morphology (Supplementary Fig. 22). SiO₂ film was deposited on top of MoO₃ layer by sputtering a 3-inch SiO₂ target with Ar plasma at a power density of 0.6 W/cm² in a radio-frequency magnetron sputtering system. NaF thin film was deposited onto the sample by heating NaF powder (Acros, 97%) loaded in a Mo boat in a high vacuum evaporator chamber (<5 × 10⁻⁵ torr). For sputtering and thermal evaporation film, film thickness was monitored by a quartz crystal microbalance and the deposition rate was maintained at 0.1 Å s⁻¹. Samples were attached to a spinning sample holder to obtain high uniformity.

**Growth of MoS₂.** High temperature growth was carried out in a 2-inch quartz tube and the temperature profile of the growth was controlled by a three-zone furnace. Sulfur powder (Aldrich, 99.98%), which was placed in an alumina crucible, and precursor sample held by a 3 × 3-cm² quartz plate were placed at the center of first and third hearing zone, respectively, as depicted in Supplementary Fig. 1c.

A 5 sccm $H_2$ and 50 sccm Ar mixed gas flow was used as carrier gas and the pressure within the quartz tube was controlled to be 30 torr. The temperature at the sample ramped up at a rate of 40 °C min$^{-1}$ to 800 °C and was held for 10 min. Sulfur vapor was introduced by ramping up the temperature at the first zone at a rate of 15 °C min$^{-1}$ and was held at the desired temperature during growth. After growth, the furnace was turned off and was fast-cooled using an industrial fan. The temperature ramping profile is shown in Supplementary Fig. 1d. For monolayer $MoS_2$ growth, A and B setpoints are reached at the same time. For bilayer $MoS_2$ growth, the B setpoint is reached 2 min later than the A setpoint.

**Device fabrication and characterization.** *p*-type heavily doped silicon wafers with 90-nm thermal oxide layers were used for back-gate FET devices. $MoS_2$ films/ isolated crystals were transferred to wafers through a conventional PMMA method. Optical lithography and oxygen plasma were used to define the $MoS_2$ strips. Then, the second lithography process defined the source-drain patterns. A 50-nm gold layer was thermally evaporated under high vacuum as the source-drain and back-gate electrodes. Before measuring electrical properties, FETs were annealed at 120 °C under a 10$^{-3}$ torr vacuum for 10 h in a probe station (Lakeshore). Gate and source-drain voltage were applied by Kethley 6487 picometers. Raman and photo-luminescence spectra were measured by a confocal system equipped with a 476-nm laser. XPS spectra were obtained using PHI VersaProbe system. Trans-mission electron analysis (Supplementary Figs. 4, 17, and 23) was performed in JEOL AEM 2010F and JEOL AEM 2100F, which was equipped with a probe-type corrector for the spherical aberration of the objective lens. Both systems were operated at 200 kV for the analysis.

## Data availability

The data that support the findings of this study are available from the corresponding author on reasonable request.

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

## Acknowledgements

We thank the Ministry of Science and Technology (MOST) Taiwan Grants MOST 107-2119-M-007-011-MY2 and MOST 106-2628-M-007-003-MY3, as well as the Academic Summit Project 107-2745-M-002-001-ASP. Financial supports by the i-MATE program in Academia Sinica, and the Center of Atomic Initiative for New Materials (AI-Mat), National Taiwan University (107L9008), from the Featured Areas Research Center Program within the framework of the Higher Education Sprout Project by the Ministry of Education (MOE) in Taiwan, are also acknowledged. The authors thank the TEM technical research services of NTU consortia of Key Technologies.

## Author contributions

M.-C.C., P.-H.H., M.-F.T., and F.-Y.L. contributed equally to this work. P.-H.H., M.-C.C., and M.-F.T. designed the method. M.-C.C., M.-F.T., F.-Y.L., H.W., P.-P.H., C.-H.C., Y.-C.Y., and H.-Y.D. helped with the CVD process and optical spectroscopy. P.-H.H. fabricated the FET devices and took the electrical measurement. C.-H.H. and J.-J.S. helped with the XPS measurement. I.-K.L., I.-T.W., and C.-Y.W. helped the TEM measurement. P.-H.H. and M.-C.C. wrote the article and discussed with C.-W.C., L.-C.C., K.-H.C., and P.-W.C.

## Competing interests

The authors declare no competing interests.
