## [Peer Review File · Nature Communications]

Reviewers' comments:

Reviewer #1 (Remarks to the Author):

In this communication, Chang et al. reported the fast growth of large-area MoS₂ single crystal continuous film via a novel self-capping vapor-liquid-solid (SCVLS) method. The as-grown millimeter-scale MoS₂ single crystal has high electron mobility up to 33 cm²/Vs on-off ration of 5×10⁸ and current density up to 230 uA/um. However, this paper needs more solid data to prove the advances of the SCVLS method. I would like the see all the following comments are well addressed before recommending publication in Nature Communications.

The following are detailed comments:

1. The SCVLS method is quite simple and universal with great potential in growing many other 2D TMDs and doped-TMDs. Here I suggest the authors demonstrated more data on these aspects. Only report the growth of millimeter MoS₂ single crystals is kind of weak for publication in Nature Communications.

2. Since the morphology of SiO₂ film changes a lot after growth, I suggest the authors demonstrate more data on the flatness of as-grown and transferred large MoS₂ flakes. Are there any cracks in the crystals which caused by the incompatible thermal expansion between MoS₂ flakes and growth substrates?

3. Uniformity of the large MoS₂ single crystals.

A. Optical uniformity: Please provide fluorescence images of the large MoS₂ flakes.

B. Electrical uniformity: It is not reasonable to choose the three far-apart regions (Edge of MoS₂ flakes). The variations of crystallinity or electrical property are always seen in the edge and center of MoS₂ flakes. More devices should be tested to clarify the electrical uniformity across the large MoS₂ flakes, from the center to edge. Here, only a few large MoS₂ flakes were studied. How about the electrical uniformity of MoS₂ flakes and films over the large, centimeter-scale substrates? More FET devices should be tested and demonstrated. In addition, when test the FETs, please apply the back-gate bias both forward and backward. I strongly suggest the authors demonstrate all the transport curves which will give the readers more information about the hysteresis and the crystallinity of MoS₂ flakes.

4. Layer-controlled growth is a very important aspect in CVD growth of 2D TMDs. Please demonstrate more data about the electrical quality of as-grown bilayer and trilayer MoS₂ flakes. This information can be a guideline for the community.

5. Page 6, after the reaction between NaF and MoO₃, it is not proper to say the residual is NaF matrix, most of the NaF may be consumed. It is better to have cross-section TEM image to evaluate the structure after MoS₂ growth. Please change the "NaF matrix" with a more proper phrase in the manuscript.

6. How to evaluate the crystallinity of MoS₂ monolayer by Raman spectroscopy, any reference? The peak separation of 19.5 cm⁻¹ can only give us the information of monolayer, not crystallinity information. It is not accurate to simply conclude high-crystallinity from Raman spectra.

7. Page 7 Line 78, Figure 3e, NO direct evidence of large-area continuous film. NO information of average grain size of 450 μm.

8. How can you control the sulfurization timing before and during the liquid's rise to the surface?

9. Page 11 Line 263. NO evidence of 100% coverage of MoS₂ monolayer. Thick layers can be seen with a high coverage on the monolayer MoS₂ film (Figure 3d).

10. Page 19 Figure 2. The insets in Figure 2b and 2c are too small to view.

11. Page 20 Figure 3g. The comparison should be more comprehensive. I would like to see more comprehensive reference data are plotted in Figure 3g, e.g. 1. Nature Communications 9(2018) 919; 2. Nanoscale 11(2019) 16122-16129, etc.

12. Page 22 Figure 5d and 5f. The insets can be presented in SI with a broad view.

13. Figure S3. It is better to have atomic-resolution STEM images for the as-grown MoS₂ crystals.

14. Figure S4. It is kind of weak to say full overage of MoS₂ film with only one photo. More solid data should be presented.

15. Figure S6b. There is no height bar for the AFM images. It is quite random format when the

scale bars were added in the figures. I suggest the authors pay more attention to these details.

16. Figure S8. What is the meaning of "MoS₂ vapor-pressure"?

17. There are several related papers were published recently, please clearly address the advancements of this work over the following works:

1. Nature Communications 9(2018) 919;
2. Nature Chemistry 11(2019) 730-736;
3. Nanoscale 11(2019) 16122-16129.

Reviewer #2 (Remarks to the Author):

The authors report a self-capping method to fabricate large grain continuous MoS₂ mono- and bi-layers. The fast growth rate and coverage of the resulting film are quite impressive. This work is interesting in general to the field and the approach adopted quite novel. However, there are still some issues to be addressed.

1, Insufficient relevant references including large grain bilayer and multilayer growth technique by using different substrate and gas ambient.

2, Details of the MoO₃ layer and NaF layer growth are lacking. Also the MoS₂ growth dependence on the NaF thickness around 15 nm needs to be provided. Why 15 nm is the optimal thickness? Will 10 nm or 20 nm be better? A step from 5 nm to 15nm to 30 nm is too large.

3, The excessive seeds and extra smaller multi-layer regions in the monolayer grains can cause performance degradation and uniformity issues in devices. How to eliminate these using the current growth technique?

4, In Fig.5, the device electric characteristics are not solid. Detailed IdVd needs to be provided and the detailed mobility calculation is also questionable. The IdVg curve shows a superlinear trend at higher V_{bg} which could cause overestimation of mobility.

5, The claims and analysis on MIT are not convincing, since this cross-over can be caused by the ohmic contact and carrier injection at various temperatures as previous reports pointed out.

6, V_{ds}=8V is considered unusually large even for a 1.48 μm gate device, and the resulting on off ratio is questionable in Fig. 5f, without multiple devices and gate leakage current.

7, The overall quality of the figures are bad, also the formality is quite careless. For example, in Fig.5f, V_d=8 is even without unit.

Reviewer #3 (Remarks to the Author):

In this work, the authors reported the controlled growth 2D MoS₂ by SCVLS method using MoO₃, NaF and S as the reaction sources, resulting in large-area single-crystal MoS₂ flakes. And the excellent electrical properties of MoS₂ FETs indicated as-synthesized MoS₂ flakes with low defects have been obtained. Meanwhile, through SCVLS method, monolayer MoS₂ can full-coverage a 3 × 3 cm² c-plane sapphire substrate with a relatively large average grain size. Therefore, I'd like to recommend its publication in Nature Communications. However, the following issues need to be clarified/addressed before acceptance.

(1) There are some white dots in the most of MoS₂ flakes, as showed in optical microscopy images (Figure 1g, 1h), suggesting that as-obtained MoS₂ nanosheet with triangular shape and 1mm in length is not single-crystal sample. And I think the white dots play an important role in growth mechanism and the quality of MoS₂ flakes, so it is necessary to characterize the white dots. In addition, the author just judges the layer number of MoS₂ flakes by optical microscopy images, which is not precise enough. I think the precise method of thickness, e.g., AFM, etc, should be shown.

(2) The authors claimed that MoO₃ vaporized and penetrated through the SiO₂ diffusion

membrane at growth temperature, so the author should explain the mechanism of SiO₂ penetrated by MoO₃, which is important for readers to understand the mechanism of the SCVLS method proposed by the authors. In addition, the authors claimed that liquid precursor rose to the NaF matrix surface through reactive digging and capillary effect, yet, generally speaking, MoO₃ vapor will break SiO₂ layer and remain holes and trenches on its surface because there is not reaction between MoO₃ and SiO₂, so after the growth finishes, the SiO₂ layer should be carefully characterized to check the surface of SiO₂ layer can deeply explaining the growth mechanism. In addition, the reaction of MoO₃ and NaF will expand NaF and provide room for liquid precursor, so it should be explored further to make clear whether the holes and trenches appear on the NaF, which is important for how the liquid precursor rise to the NaF matrix surface.

(3) For FET, the authors fabricated and measured 18 FETs, which show 33 cm²V⁻¹s⁻¹ of average mobility and 5*10⁸ of on/off ratio, but some things as followed are indistinct: 1) the mobility, on/off ratio, image, thickness of each device were not found in main text and SI, pls give AFM tests and some channel materials may be not monolayer because some white dots are found in 1-mm MoS₂ crystal ; 2) the author should add one column in the Table 1, which indicates the thicknesses of the measured samples because the comparison of FET properties of different thicknesses MoS₂ is meaningless.

Response letter for *Fast Growth of Millimeter-Size Two-Dimensional MoS₂ through a Self-Capping Vapor-Liquid-Solid Method.*

We have considered the comments of all referees and have modified the manuscript accordingly. Below, we have addressed each of the referees' comments point-by-point.

Reviewer #1 (Remarks to the Author):

In this communication, Chang et al. reported the fast growth of large-area MoS₂ single crystal continuous film via a novel self-capping vapor-liquid-solid (SCVLS) method. The as-grown millimeter-scale MoS₂ single crystal has high electron mobility up to 33 cm²/Vs on-off ration of 5×10⁸ and current density up to 230 uA/um. However, this paper needs more solid data to prove the advances of the SCVLS method. I would like the see all the following comments are well addressed before recommending publication in Nature Communications.

The following are detailed comments:

1. The SCVLS method is quite simple and universal with great potential in growing many other 2D TMDs and doped-TMDs. Here I suggest the authors demonstrated more data on these aspects. Only report the growth of millimeter MoS₂ single crystals is kind of weak for publication in Nature Communications.

Ans: We would like to thank the reviewer for the comments and suggestions. We also think the SCVLS has great potential for growing other 2D, doped 2D and for alloying 2D. We are trying to add new precursors in our ALD system to grow WO₃ and Nb₂O₅. Here we provide some structures that are designed for doped TMD and alloyed 2D as shown in Fig. R1. In addition, another importance of this work is that the new SCVLS mechanism can grow large-grain continuous film, which cannot be achieved in conventional gas phase CVD process as shown in Fig 3g. We also change the title of manuscript to "Fast Growth of Large-Grain, Continuous MoS₂ Films through a Self-Capping Vapor-Liquid-Solid Method".

Figure R1. Application of SCVLS method. (a) Replace MoO₃ and S precursors by WO₃ and Se for WS₂ and WSe₂. (b) Insert a Nb₂O₅ layer for Nb-doped MoS₂. (c) Insert a WO₃ layer for Mo_xW_{1-x}S₂ alloy.

2. Since the morphology of SiO₂ film changes a lot after growth, I suggest the authors demonstrate more data on the flatness of as-grown and transferred large MoS₂ flakes. Are there any cracks in the crystals which caused by the incompatible thermal expansion between MoS₂ flakes and growth substrates?

Ans: Fig. S12a and b are the optical and AFM images of the as-grown MoS₂ on sapphire. The deliquescence of the bottom NaF matrix and non-fully sulfurized products is very fast. The AFM image (Fig. S12b) of the as-grown MoS₂ is comparatively rougher than the as-transferred MoS₂ (Fig. S12d) because of the deliquescence of the substrate (however, it takes at least 10 minutes to get an AFM image so we could not avoid the deliquescence). For transferred MoS₂, we encapsulate the as-grown MoS₂ by PMMA as soon as we take it out from the furnace, preventing the deliquescence to happen. The transferred MoS₂ on silicon substrate shows very smooth surface with average roughness around 0.19 nm, which is comparable to previous reported values. This result indicates there is no obvious incompatible thermal expansion or strain between MoS₂ and the substrate. The good electrical properties of SCVLS MoS₂ also confirm the high quality of the transferred MoS₂.

Figure S12. AFM analysis of SCVLS MoS₂ (a) Optical and (b) AFM images of the as-grown MoS₂. (c) Optical and AFM images of the transferred MoS₂ on silica. All scale bars in the images are 20 μm. The average roughness of (b) and (d) are 0.29 and 0.19 nm, respectively.

3. Uniformity of the large MoS₂ single crystals.

A. Optical uniformity: Please provide fluorescence images of the large MoS₂ flakes.

B. Electrical uniformity: It is not reasonable to choose the three far-apart regions (Edge of MoS₂ flakes). The variations of crystallinity or electrical property are always seen in the edge and center of MoS₂ flakes. More devices should be tested to clarify the electrical uniformity across the large MoS₂ flakes, from the center to edge. Here, only a few large MoS₂ flakes were studied. How about the electrical uniformity of MoS₂ flakes and films over the large, centimeter-scale substrates? More FET devices should be tested and demonstrated. In addition, when test the FETs, please apply the back-gate bias both forward and backward. I strongly suggest the authors demonstrate all the transport curves which will give the readers more information about the hysteresis and the crystallinity of MoS₂ flakes.

Ans: Fig. S13 is the photoluminescence mapping of the as-grown MoS₂ (the

limitation of our mapping system is $75\text{ }\mu\text{m}\times 75\text{ }\mu\text{m}$), the uniform PL signal confirms the uniformity of SCVLS MoS_2 . For electrical measurement, we provided the data for the forward and backward scans as shown in Fig. 5(c). The hysteresis of our device is very small, which indicates the low defect-induced trap and impurity density of the SCVLS MoS_2 . For uniformity, we transferred a $1.5\text{ cm}\times 1.5\text{ cm}$ continuous MoS_2 film to a $2.5\text{ cm}\times 2.5\text{ cm}$ silicon wafer. Hundreds of devices were fabricated as shown in figure 6 (a)-(c). Nearly 90% of devices are pure monolayer devices. 4% are pure bilayer and the other 6% are few-layer and mono-few (mixed) layer devices. In figure 6 (d), we show the gate dependent conductance of randomly selected 100 devices cross the whole $1.5\text{ cm}\times 1.5\text{ cm}$. The pure monolayer shows great uniformity with narrow distribution of mobility ($34\pm 7\text{ cm}^2\text{V}^{-1}\text{S}^{-1}$) and threshold voltage ($4.9\pm 2.3\text{ V}$). For bilayer and multilayer devices, higher mobility and smaller threshold voltage were observed in the figures. For the monolayer with few-layer flake on top or monolayer-few layer junction devices, the mobility and V_{th} variation are larger, but the performance is still comparable to the monolayer devices because of a large-grain and continuous monolayer underneath (Figure S19). Having high-performance across a large area is crucial for the practical application.

Figure S13. (a) Photoluminescence mapping image of the as-grown MoS_2 on sapphire. (b) Photoluminescence spectrum.

Figure 6. Transport properties of the large-grain, continuous film. (a) Photo-image of the as-fabricated FET devices cross a 1.5 cm \times 1.5 cm region. (b) Optical image of FET devices, the scale bar is 150 μ m. (c) Gate-dependent conductance of the devices across the large area.

Figure S19. Monolayer device with a top flake (a) Optical image of a monolayer device with a top flake. The scale bar is 5 μ m. (b) Transport properties of the device with pure monolayer and monolayer with a top flake.

4. Layer-controlled growth is a very important aspect in CVD growth of 2D TMDs. Please demonstrate more data about the electrical quality of as-grown bilayer and trilayer MoS₂ flakes. This information can be a guideline for the community.

Ans: Fig. 5(c) is the gate-dependent conductance of bilayer and monolayer MoS₂. The bilayer MoS₂ has a lower threshold voltage because of their higher intrinsic carrier concentration. The mobility of bilayer (49 $\text{cm}^2\text{V}^{-1}\text{s}^{-1}$) is also higher than the monolayer (33 $\text{cm}^2\text{V}^{-1}\text{s}^{-1}$), which is comparable to previous report of high-quality

bilayer MoS₂. For trilayer MoS₂, we did not provide its transport because it is still difficult to get large and pure trilayer crystals (there is still some non-uniform region in Fig. 4e, so we only claimed the large bilayer crystals in the manuscript). The layer-control growth is very sensitive to sulfurization timing as we described in the manuscript. More sophisticated sulfurization technique such as using H₂S_(g) as sulfur source is expected to control the layer number of SCVLS MoS₂ better.

Figure 5. Transport properties of the MoS₂ grown through SCVLS method. (a) Schematic image of a back-gate monolayer MoS₂ transistor. (b) Optical image of a monolayer and bilayer FET device. Both scale bars are 5 μm. (c) Gate-dependent conductance of corresponding to mono- and bilayer devices shown in (b).

5. Page 6, after the reaction between NaF and MoO₃, it is not proper to say the residual is NaF matrix, most of the NaF may be consumed. It is better to have cross-section TEM image to evaluate the structure after MoS₂ growth. Please change the “NaF matrix” with a more proper phrase in the manuscript.

Ans: Here we would like to explain with more details that keeping the term “NaF-matrix” is appropriate as followings. In SCVLS reaction, the NaF is used as a reagent and a substrate for growing MoS₂, so there should be excess NaF in the system. We use 20 nm NaF and 3-7 nm MoS₂ precursor for growing MoS₂ films. Considering the density and the molecule weight of NaF and MoO₃, the amount of NaF deposited is much more than the amount needed for the eutectic reaction. Therefore, we would have a NaF matrix with non-fully sulfurized products as shown in Figure 2a-c, which are water soluble, after growth. They can be easily washed away by water (Figure S6a and b), leaving porous SiO₂ surface as shown in Fig. S6c. This phenomenon proves the non-fully sulfurized residuals (MoO_x, MoS_xO_{2-x}) are inside the water-soluble matrix, which includes NaF and Na₂Mo₂O₇. The porous SiO₂ also proves our mechanism that the MoO₃ vapor breaks the original smooth SiO₂ layer (Fig. S6d) so that it can gradually react with NaF as we proposed. In addition,

the change in morphology (Fig. S6e) of the sample after 1 week caused by the deliquescence also proves the MoS₂ is grown on a NaF matrix with non-fully sulfurized products, confirming our mechanism. We wanted to show the path within the NaF matrix by TEM cross-section, but we met difficulties when making the TEM sample by FIB because of the low conductivity of our sample and the drastic difference in hardness between sapphire and the rest of the film.

Figure S6. (a) Schematic and (b) photo image of the as-grown MoS₂ after partially immersed in water. (c) AFM image of the red square region in (b). (d) AFM image of the as-sputtered SiO₂ layer. (e) optical image of blue square region in (b) after one week in the air.

6. How to evaluate the crystallinity of MoS₂ monolayer by Raman spectroscopy, any reference? The peak separation of 19.5 cm⁻¹ can only give us the information of monolayer, not crystallinity information. It is not accurate to simply conclude high-crystallinity from Raman spectra.

Ans: We used Raman spectroscopy to show that there is no crystalline MoS₂ and other materials at places without MoS₂ coverage, not to show that the MoS₂ has good crystallinity. The crystallinity of MoS₂ is mainly shown by TEM diffraction pattern in Fig. S3. The small full width at half maximum in the Raman spectrum is a supporting evidence of good crystallinity.

7. Page 7 Line 78, Figure 3e, NO direct evidence of large-area continuous film. NO information of average grain size of 450 μm.

Ans: Fig. 3d shows that the surface is fully covered by at least a monolayer of MoS₂

with some flakes on top of the first layer. Fig. 3e shows the growth result under the same condition as in Fig. 3d, but the growth time is shortened to 1 minute (10 minutes for Fig 3d). Since the growth is really fast (shown by achieving high coverage within one minute of growth), we can expect that there will be few new nucleus formed before the MoS₂ monolayer covers the surface. Therefore, we measured the grain size of the sample with reduced growth time and scaled it with coverage to obtain the grain size of the full-coverage film, which we estimated to be 450 μm.

8. How can you control the sulfurization timing before and during the liquid's rise to the surface?

Ans: We control the timing of introducing sulfur by changing the timing we start to heat the sulfur powder. This is mentioned in Fig. S1 and experimental section.

9. Page 11 Line 263. NO evidence of 100% coverage of MoS₂ monolayer. Thick layers can be seen with a high coverage on the monolayer MoS₂ film (Figure 3d).

Ans: As stated in our answer to question (7), we have shown the full coverage of MoS₂ in Fig. 3d. The non-uniformity of the MoS₂ grown by this method is the main issue. Currently, we suggest that this issue can be solved by the layer-resolved 2D material splitting technique developed by Shim, J. et al. (ref. 34), wherein the monolayer was peeled off by nickel film as shown in Figure R2. Meanwhile, we are working on improving the uniformity of this method.

Figure R2. Schematic image of the layer-resolved 2D material splitting technique.

10. Page 19 Figure 2. The insets in Figure 2b and 2c are too small to view.

Ans: We have enlarged the inset as shown in new Figure 2.

11. Page 20 Figure 3g. The comparison should be more comprehensive. I would like to see more comprehensive reference data are plotted in Figure 3g, e.g. 1. Nature Communications 9(2018) 919; 2. Nanoscale 11(2019) 16122-16129, etc.

Ans: Among the references suggested by the reviewer, we have now included the one related to material growth (Nanoscale 2019, 11, 16122-16129) in Fig. 3g, while the other is not a paper about materials growth). In addition, we show the result of another gas phase reference in Fig. 3g for more comparisons. Clearly, the average grain size of gas phase reactions drops dramatically when more precursor is used for continuous film, which is the result of high nucleation density. As for another VLS paper, the average grain size is 100 μm and did not have data of different coverages. For our SCVLS, the self-capping effect constrains the nucleation region, so the large-grain and continuous film can be achieved.

Figure 3 (g) Comparison of SCVLS, VLS and gas-phase CVD. SCVLS reaction enables a relatively large grain size when a full-coverage film was achieved.

12. Page 22 Figure 5d and 5f. The insets can be presented in SI with a broad view.

Ans: We have rearranged Figure 5 and put the short channel device image in Figure S18.

13. Figure S3. It is better to have atomic-resolution STEM images for the as-grown MoS₂ crystals.

Ans: Fig. S14 is the atomic-resolution STEM image of the SCVLS MoS₂ taken under 200 kV electron beam (JEOL AEM 2010F). The hexagonal lattice shown in STEM pattern shows the basal plane (001) of MoS₂. Because the contrast of STEM image is proportional to the atomic number, the contrast of the Mo atom is much higher than the S atom. The clear atomic contrast from Fig. S14 is from the Mo atom. The lattice constant of the SCVLS MoS₂ can be directly measured as 0.306 nm. This image is similar to previously reported image taken in the 200 kV condition (*Scientific Reports*, **2013**, 3, 1866).

Figure S14. Atomic-resolution high-angle annular dark-field scanning transmission electron microscopy (HAADF-STEM) image for the SCVLS MoS₂

14. Figure S4. It is kind of weak to say full overage of MoS₂ film with only one photo. More solid data should be presented.

Ans: More optical microscopy images are included in Fig. S4 to show the full coverage of MoS₂.

15. Figure S6b. There is no height bar for the AFM images. It is quite random format when the scale bars were added in the figures. I suggest the authors pay more attention to these details.

Ans: We thank the reviewer's reminder. We added the scale bar in Fig. S6c.

16. Figure S8. What is the meaning of "MoS₂ vapor-pressure"?

Ans: We are sorry for the typo. We have replaced it with "MoO₃" vapor pressure.

17. There are several related papers were published recently, please clearly address the advancements of this work over the following works:

1. Nature Communications 9(2018) 919;
2. Nature Chemistry 11(2019) 730-736;
3. Nanoscale 11(2019) 16122-16129.

Ans: We are grateful for the reviewer's suggestion. However, we found that only the third paper is related to 2D growth (the first two seem to be non-related topics); therefore, we include this and another recent paper (Scientific reports, 2015, 5, 18596) in our revised list of references and also compare the coverage and grain in Fig 3(g). Clearly, the SCVLS gives the largest average grain size of the continuous film because of the new self-capping mechanism. This large-grain, continuous MoS₂ film also exhibits great transport behavior across a centimeter-scale area as shown in Fig. 6. In addition, the control of layer number is another advantage of using SCVLS.

Reviewer #2 (Remarks to the Author):

The authors report a self-capping method to fabricate large grain continuous MoS₂ mono- and bi-layers. The fast growth rate and coverage of the resulting film are quite impressive. This work is interesting in general to the field and the approach adopted quite novel. However, there are still some issues to be addressed.

1, Insufficient relevant references including large grain bilayer and multilayer growth technique by using different substrate and gas ambient.

Ans: We would like to thank the reviewer for their comments and suggestions. We added the following references related to layer-controlled growth in the revised manuscript.

1. Advanced Materials, 2017, 29, 1604540 (multilayer)
2. Nature Communications, 2019, 10, 598 (bilayer)
3. Nature communications, 2018, 9, 4778 (bilayer)
4. 2D Materials, 2019, 6, 025030 (mono-multilayer)

2, Details of the MoO₃ layer and NaF layer growth are lacking. Also the MoS₂ growth dependence on the NaF thickness around 15 nm needs to be provided. Why 15 nm is the optimal thickness? Will 10 nm or 20 nm be better? A step from 5 nm to 15nm to 30 nm is too large.

Ans: Thanks for reviewer's suggestion. We have provided more details of the whole

SCVLS in the revised manuscript. MoO₃ film with well-controlled thickness was grown on top of sapphire substrates with a homemade plasma-enhanced atomic layer deposition system using Mo(CO)₆ as precursor and oxygen plasma as the oxidation method. For each deposition cycle, a Mo(CO)₆ precursor pulse is provided into the chamber, then the excess precursor is purged away by argon, and finally using oxygen plasma (up to 200W) to oxidize the precursor and deposit uniform MoO₃ film. SiO₂ film was deposited on top of MoO₃ layer by sputtering a 3-inch SiO₂ target with Ar plasma at a power density of 0.6 W/cm² in a radio-frequency magnetron sputtering system. NaF thin film was deposited onto the sample by heating NaF powder (Acros, 97%) loaded in a Mo boat in a high vacuum evaporator chamber (<5×10⁻⁵ torr). For sputtering and thermal evaporation film, film thickness was monitored by a quartz crystal microbalance and the deposition rate was maintained at 0.1 Å s⁻¹. Samples are attached to a spinning sample holder to obtain high uniformity.

As for the NaF thickness dependency, here we provide more data with different NaF thickness as shown in Fig. S7. The best NaF thickness for largest average grain is set around 15-20 nm. MoS₂ grown with 30 nm NaF have bad morphology and it is difficult to measure its grain size, so the grain size is roughly estimated.

Figure S7. NaF thickness effect. Optical images of SCVLS MoS₂ with a (a) 5 nm, (b) 10 nm, (c) 15 nm, (d) 20 nm, and (e) 30 nm NaF layer. Scale bars are 500 μm in (a)-(e). The average grain size of MoS₂ grown with different NaF thickness is shown in (f). The grain size of MoS₂ increases firstly as NaF gets thicker because the thicker NaF reduces the liquid droplet density during the growth stage. However, if the NaF is too thick, it is difficult for liquid to go up to surface. With a lot of liquid at the bottom, the whole structure becomes very unstable, so the morphology of MoS₂ becomes very non-uniform as shown in (e). Here, 15~20 nm is found to be the best thickness for growing large MoS₂ crystals.

3, The excessive seeds and extra smaller multi-layer regions in the monolayer grains can cause performance degradation and uniformity issues in devices. How to eliminate these using the current growth technique?

Ans: The morphology of the thick or non-uniform layer is due to different sulfurization timing and the amount of sulfur vapor. If the sulfurization occurs late and the amount of sulfur pressure is suitable, the sulfurization of the large liquid drop on NaF surface results in bilayer or trilayer structure as shown in Figure 4. If the sulfur vapor pressure is too high, the large droplet will quickly be sulfurized and solidified, forming thick flakes on top of the monolayer as shown in Fig. 3d. Therefore, to eliminate this nonuniformity, the most feasible method is to precisely control the amount and introducing timing of S source. Compared with sulfur powder, gas phase H₂S is expected to be much easier to control the amount and the introducing timing of sulfur, which has great potential to improve the uniformity of SCVLS method.

4, In Fig.5, the device electric characteristics are not solid. Detailed IdVd needs to be provided and the detailed mobility calculation is also questionable. The IdVg curve shows a superlinear trend at higher Vbg which could cause overestimation of mobility.

Ans: Here we have provided the I_d-V_d curve of the device with 1.48 μm channel length as shown in Fig. 5e and R3(a). For mobility, instead of using the peak value of $\frac{\partial \sigma}{\partial V_G}$ for calculating mobility (which may overestimate the mobility of devices), we use the average slope of the 30V-40V region as shown in Fig. R2, which shows an excellent linear fit to the as-measured curve. In addition, compared with many CVD papers that fabricate devices without etching triangle crystals into ribbons with a precise width, we believe our mobility calculation is more precise and solid.

Figure R3. (a) I_d - V_d characteristic of a short channel (1.48 μm) device. (b) Gate-dependent conductance of the device. (c) Linear fitting for the mobility calculation.

5, The claims and analysis on MIT are not convincing, since this cross-over can be caused by the ohmic contact and carrier injection at various temperatures as previous reports pointed out.

Ans: The reviewer's comment is well taken. Yet, we would like to claim the importance of MIT transition here does have a solid ground. Generally, the device performance is strongly influenced by the contact property and the intrinsic mobility of channel material. For a clear MIT transition, both high-mobility channel and good Ohmic contact are needed. For the contact issue, if it is a Schottky contact, the contact resistance increases at lower temperatures because of thermionic emission, which pushes the MIT point to high gate voltage region or even invisible at a reasonable gating range. Considering the material itself, MIT occurs when the critical channel conductance is approximately one quantum conductivity (e^2/h) according to the Ioffe-Regel criterion as we described in our manuscript. The conductivity of the materials is decided by $\sigma = nq\mu$, where n is the carrier density of material, μ is the mobility of material. For MIT condition, $\sigma_{MIT} = n_{MIT}q\mu$, if the mobility of material is high, the n_{MIT} would be lower. Therefore, the low n_{MIT} would occur in the high mobility device, which is also demonstrated in the previous reports [ref 45-47]. Compared with many reports that just show an I_d - V_G curve, we think it is a better way to demonstrate the high-quality of our SCVLS MoS_2 and the good contact of our device.

6, $V_{ds} = 8\text{V}$ is considered unusually large even for a 1.48 μm gate device, and the resulting on off ratio is questionable in Fig. 5f, without multiple devices and gate leakage current.

Ans: For high-field transport behavior of MoS_2 such as maximum current density or saturation velocity, it is common to have electric field larger than 5 $\text{V}/\mu\text{m}$. Here we included an example from *ACS Nano* 2015, 9, 8, 7904-7912. They use 8V in a 0.8 μm (10 $\text{V}/\mu\text{m}$) channel to get the maximum current density of their channel as shown in Fig. R4. We also provided the I_d - V_d curve of our device in Fig. 5e and R5a. Excellent linear behavior in the low field indicates good mobility and the observation of slight saturation at 8 V is similar to what is observed in the previous work with high-quality exfoliated MoS_2 devices (Fig. R4c). The leakage current in our system is around $5 \times 10^{-12} \sim 2 \times 10^{-11} \text{A}$ depends on the applied gate voltages and devices. We also provided another device curve in Fig. R5b, which have similar on-off ratio around

$1 \times 10^8 \sim 5 \times 10^8$.

Figure R4. High-field characteristics of back-gated monolayer MoS₂ FETs from ACS Nano 2015, 9, 8, 7904-7912.

Figure R5. (a) I_d - V_d characteristic of a short channel (1.48 μm) device. (b) Gate-dependent current density of two short devices.

7, The overall quality of the figures are bad, also the formality is quite careless. For example, in Fig.5f, $V_d=8$ is even without unit.

Ans: We thank the reviewer for their reminder. We have modified and improved the quality of figures.

Reviewer #3 (Remarks to the Author):

In this work, the authors reported the controlled growth 2D MoS₂ by SCVLS method using MoO₃, NaF and S as the reaction sources, resulting in large-area single-crystal MoS₂ flakes. And the excellent electrical properties of MoS₂ FETs indicated as-synthesized MoS₂ flakes with low defects have been obtained. Meanwhile, through SCVLS method, monolayer MoS₂ can full-coverage a $3 \times 3 \text{ cm}^2$ c-plane

sapphire substrate with a relatively large average grain size. Therefore, I'd like to recommend its publication in Nature Communications. However, the following issues need to be clarified/addressed before acceptance.

(1) There are some white dots in the most of MoS₂ flakes, as showed in optical microscopy images (Figure 1g, 1h), suggesting that as-obtained MoS₂ nanosheet with triangular shape and 1mm in length is not single-crystal sample. And I think the white dots play an important role in growth mechanism and the quality of MoS₂ flakes, so it is necessary to characterize the white dots. In addition, the author just judges the layer number of MoS₂ flakes by optical microscopy images, which is not precise enough. I think the precise method of thickness, e.g., AFM, etc, should be shown.

Ans: We would like to thank the reviewer for their comments and suggestions. Raman and AFM images of the white dot are included in Fig S15. The big white dot is thick MoS₂ having thickness around 20-35 nm. We speculate these white dots come from the liquid precursor that overflowed to the top during the growth of MoS₂ and eventually sulfurized into MoS₂ flake on top of the monolayer. However, this process does not interrupt the proposed self-capping growth process beneath it and we have shown that our ~1mm grain is a single crystal by TEM diffraction images in Figure S3. For thickness confirmation, we included an additional AFM image in Figure S17 to confirm the thickness of mono- and bilayer MoS₂. In addition, we also have Raman and photoluminescence data shown in Fig. 4f and g that can confirm the layer number of MoS₂ flakes.

Figure S15. Characterizations of the white dot (flake) on a monolayer grain. (a) Optical image of the white dot. (b) AFM image of the white dots. Height profiles is shown in (c). (d) Raman spectrum of the white dot.

Figure S16. (a) Optical image of a bilayer flake. The scale bar is 10 μm (b) AFM image of the bilayer MoS_2 flake. (c) Height profile of the dash line marked in (b).

(2) The authors claimed that MoO_3 vaporized and penetrated through the SiO_2 diffusion membrane at growth temperature, so the author should explain the mechanism of SiO_2 penetrated by MoO_3 , which is important for readers to understand the mechanism of the SCVLS method proposed by the authors. In addition, the authors claimed that liquid precursor rose to the NaF matrix surface through reactive digging and capillary effect, yet, generally speaking, MoO_3 vapor

will break SiO₂ layer and remain holes and trenches on its surface because there is not reaction between MoO₃ and SiO₂, so after the growth finishes, the SiO₂ layer should be carefully characterized to check the surface of SiO₂ layer can deeply explaining the growth mechanism. In addition, the reaction of MoO₃ and NaF will expand NaF and provide room for liquid precursor, so it should be explored further to make clear whether the holes and trenches appear on the NaF, which is important for how the liquid precursor rise to the NaF matrix surface.

Ans: The SiO₂ surface after growth was studied by AFM, shown in Fig. S6c. The AFM image shows some ~200 nm pinholes that we believe are the spots where the MoO₃ vapor breaks the thin SiO₂ layer.

Figure S6. Analysis of growth. In our SCVLS reaction, the NaF is used as a reagent and a substrate for growing MoS₂, so the system must have excess NaF. We use 20 nm NaF and 3-7 nm MoS₂ precursor for growing MoS₂ films. Considering the density and the molecule weight of NaF and MoO₃, the amount of NaF deposited is much more than the amount needed for the eutectic reaction. Therefore, there will be a NaF matrix with non-fully sulfurized products as shown in Figure 2(a) and (b), which are water soluble, after growth. They can be easily washed away by water (Figure S6a and b), leaving a porous SiO₂ surface as shown in Figure S6c. This phenomenon, together with the depth profile XPS results, shows that the non-fully sulfurized residuals (MoO_x, MoS_xO_{2-x}) are buried in the water-soluble matrix, including NaF and Na₂Mo₂O₇. The porous SiO₂ also proves that the MoO₃ vapor breaks the original smooth SiO₂ layer (Figure S6d) so that it can gradually react with NaF as in our proposed mechanism. In addition, the morphology of the sample changes (Figure S6e) after 1 week because of the deliquescence. This proves that the MoS₂ is grown

on a NaF matrix with non-fully sulfurized products and confirms our mechanism.

(3) For FET, the authors fabricated and measured 18 FETs, which show $33 \text{ cm}^2\text{V}^{-1}\text{s}^{-1}$ of average mobility and 5×10^8 of on/off ratio, but some things as followed are indistinct: 1) the mobility, on/off ratio, image, thickness of each device were not found in main text and SI, pls give AFM tests and some channel materials may be not monolayer because some white dots are found in 1-mm MoS₂ crystal ; 2) the author should add one column in the Table 1, which indicates the thicknesses of the measured samples because the comparison of FET properties of different thicknesses MoS₂ is meaningless.

Ans: The mobility, on-off ratio and device image were already included in the manuscript. The additional AFM image of the channel is included in Fig. S17. The photoluminescence and Raman data in the manuscript also confirm the layer number of MoS₂. Every data in the table are “pure monolayer” MoS₂ TFT devices. We also measured a hundred devices across a 1.5 cm x 1.5 cm region. Nearly 90% of devices are pure monolayer devices. 4% are pure bilayer and the other 6% are few-layer and mono-few-layer (mixed) devices. In figure 6 (c), we show the gate dependent conductance of 100 devices cross the whole 1.5 cm X 1.5 cm. The pure monolayer shows great uniformity with narrow distribution of mobility ($34 \pm 7 \text{ cm}^2\text{V}^{-1}\text{s}^{-1}$) and threshold voltage ($4.9 \pm 2.3 \text{ V}$). For bilayer and multilayer devices, higher mobility and smaller threshold voltage were observed in the figures. For the monolayer with few-layer flake on top or monolayer-few layer junction devices, the mobility and V_{th} variation are larger, but the performance is still comparable to the monolayer devices because of a large-grain and continuous monolayer underneath. Fig. S19 is an example of a monolayer device with a flake on top. The performance of such device is similar to pure monolayer MoS₂ (only with a slight shift of V_{th}).

Figure S17. AFM measurement of a monolayer FET device. (a) Optical and (b) AFM

images of the monolayer MoS₂ device. The jagged edges of electrodes are due to the metal liftoff process. The scale bars in (a) and (b) are 5 μm. (c) Height profile of the monolayer MoS₂.

Figure 6. Transport properties of the large-grain, continuous film. (a) Photo-image of the as-fabricated FET devices cross a 1.5 cm × 1.5 cm region. (b) Optical image of FET devices, the scale bar is 150 μm. (c) Gate-dependent conductance of the devices across the large area.

Figure S19. Monolayer device with a top flake (a) Optical image of a monolayer device with a top flake. The scale bar is 5 μm. (b) Transport properties of the devices with pure monolayer and monolayer with a top flake.

Reviewers' comments:

Reviewer #1 (Remarks to the Author):

In this revised manuscript, several comments are still not clearly answered. There are so many hypotheses in the growth mechanism which make most of the discussion superficial. The overall image quality is still poor and below the standard. The illustrations need to be improved. The optical images should have optimal brightness and contrast, otherwise, they are hard to view. I insisted the authors take all the comments seriously, improve the quality of figures, do additional experiments and provide solid data to support their claims. Therefore, I will not recommend publication in Nature Communications after all the comments are well addressed.

1. The authors change the title to "Fast Growth ..." so how fast? Is it comparable to the previous record: 300 $\mu\text{m}/\text{min}$? See reference, Yuping Shi et al Na-assisted fast growth of large single-crystal MoS₂ on sapphire, Nanotechnology 2019 30 034002. I would like to see a more clear discussion on "the fast growth".

Abstract:

2. "Conventional chemical vapor deposition methods for two-dimensional materials generally use an extremely small amount of precursor to render large single-crystal flakes, which inevitably causes low coverage of the materials on the substrate." The judgment is only concluded from one reference, I didn't think it can be generalized to "conventional chemical vapor deposition method".

3. "An intermediate liquid phase Na₂Mo₂O₇ is formed through a eutectic reaction,". Please revise this sentence to "An intermediate liquid phase Na₂Mo₂O₇ is formed through a eutectic reaction of MoO₃ and NaF,".

4. "The field-effect transistors fabricated from the full-coverage films show high mobility (33 and 49 $\text{cm}^2\text{V}^{-1}\text{s}^{-1}$ for the mono and bilayer regions)". It should be the average mobility and the value 33 cm^2/Vs is conflict to the data present in Page 11: "90% of devices have pure monolayer channel and show high mobilities ($34 \pm 7 \text{ cm}^2 \text{ V}^{-1}\text{s}^{-1}$) with a small variation of V_{th} ($4.9 \pm 2.3 \text{ V}$)".

Introduction:

5. Page 4: "However, because of the relatively low melting temperature of NaCl and rapid eutectic reaction, this method can only generate MoS₂ nanoribbons, which considerably limits its application." This comment totally misunderstands the paper. The reason for growing 1D MoS₂ nanoribbon only because of the low wettability between the growth substrate and the liquid droplets, the liquids prefer to keep spheres during the sulfurization process. In the reference 31, wafer-scale 2D monolayer MoS₂ films can be grown with these liquid precursors, e.g., Na₂MoO₄.

Results:

Material synthesis and growth mechanism.

One obvious drawback in this part is that the authors use too many hypotheses and schematic illustrations instead of direct experimental data. That makes all the discussions weak and no roots.

6. Page 4: "Figure 1a and S1 show that a smooth MoO₃ layer was grown on c-plane sapphire through plasma-enhanced atomic layer deposition." There is no data indicating a smooth MoO₃ layer, only a schematic illustration.

7. Page: "Figure S4 shows a full-coverage monolayer MoS₂ film grown on a sapphire of $3 \times 3 \text{ cm}^2$ (size was only limited by the CVD tube size)". It is not a fully-coverage monolayer MoS₂ film. It

also contains voids, thick layers. Please mention the ratio of monolayer, thick-layers, and voids.

Controlling the coverage and thickness of MoS₂.

8. The understanding of the growth mechanism can be wrong. We need more direct evidence, just propose a schematic illustration is meaningless.

Here are some core questions must be answered with direct evidence. The authors mentioned the later introduction of the sulfur, the Na₂MoO₇ will grow up to large droplet and lead to the growth of bilayer MoS₂. So please do more controlled experiments to prove it.

In control experiments without introducing sulfur, what are the area-density and size of the Na₂MoO₇ particles with different annealing times at the growth temperature?

9. Page 7: "The average grain size of the full-coverage film was approximately 450 μm" The authors estimate the nucleation only happened in the first 1 minute, but the truth is the nucleation of MoS₂ can happen at the whole growth process. And the grain size of approximately 450 μm is strongly against the data presents in Figure S4.

10. Page 8: "Although there are some small islands on the film (Figure 3d)" I don't think thick layers with a dimension of hundreds of micrometers is "small". From Figure 3d, we can see the thick layers are very large with a high ratio in the as-grown film. This non-uniformity will cause problems when integrating it into electronic devices.

11. Page 8: "the bottom continuous film can be peeled off and transferred to various substrates with the recently developed layer-resolved 2D material splitting technique³⁴." If the authors think the method reported in Ref. 34 is applicable to their samples, please demonstrated it with direct efforts. This kind of proposal is meaningless.

Technical issue:

12. The mention of supplementary figures in manuscript is in quite random order. No mention of Figure S12-18 in the manuscript.

Supplementary Information

13. Figure S1c: It should be "temperature ramping profile".

14. Figure S6: Please change "3-7 nm MoS₂ precursor" to "3-7 nm MoO₃"

15. Figure S7: What are the thickness of SiO₂ and MoO₃ layers?

16. Figure S8: What are the thickness of SiO₂, MoO₃ and NaF layers?

17. Figure S9: What are the thickness of NaF and MoO₃ layers?

Rebuttal Letter

18. Question 1: In Figure R1, the authors proposed the growth of tungsten-based TMDs and TMD alloys, but we DO want to see more experimental data on these aspects.

19. Question 2: The AFM results (Figure S12) only show small MoS₂ flakes, but cracks can be seen in large-grain MoS₂ films as shown in Figure S4.

20. Question 11 and 17 references

1. Batch production of 6-inch uniform monolayer molybdenum disulfide catalyzed by sodium in glass, Nature Communications 9(2018) 919. This is very important progress in CVD growth of large-area (6-inch) MoS₂ film with a grain size of 400 μm in 8-min growth on a cheap glass substrate. This paper shows many advantages over the current paper. It must be included in Table 1 and Figure 3g. Intentionally overlook this important literature is unacceptable.

2. Kinetic modulation of graphene growth by fluorine through spatially confined decomposition of metal fluorides. Nature Chemistry 11(2019) 730-736. This is the first research paper that uses fluoride as growth promoters. It works well not only for the growth of TMDs but also for graphene and h-BNs. It is an important progress in the "salt-assisted growth of 2D materials". It is also

strongly related to this work, using NaF as a growth promoter.

3. Nanoscale 11(2019) 16122-16129. The citation in Reference should change from "Li, S. et al. Wafer-Scale and Deterministic Patterned Growth of Monolayer MoS₂ via Vapor-Liquid-Solid Method. arXiv Prepr. arXiv1906.05436 (2019)." to "Li, S. et al. Wafer-Scale and Deterministic Patterned Growth of Monolayer MoS₂ via Vapor-Liquid-Solid Method. Nanoscale 11 16122-16129

Reviewer #2 (Remarks to the Author):

After reading the response letter thoroughly, as well as the revised manuscript, the reviewer feels this manuscript still falls short to the standard in nature communications for the following reasons.

1, The VLS method has been used previously.

2, The self-capping effect is not well controlled, as the multilayer region cannot be completely suppressed.

3, The monolayer quality has only incremental improvement over previous results in terms of mobility or current.

4, The wafer scale result is not convincing with very few surface and material characterization. Previous work (ACS Nano 2017) has demonstrated even better growth results.

5, The overall figure quality and data presentation is of low-quality, far below the standard of nature communications. The reviewer has raised this issue in the first round process and the quality has not been improved.

Overall, this work lacks the novelty, significance and quality to be accepted in Nature Communications.

Reviewer #3 (Remarks to the Author):

The questions have been addressed by the point to point responses of the author and I agree that this work is published in Nature Communications.

Reviewer #1 (Remarks to the Author):

1. The authors change the title to “Fast Growth ...” so how fast? Is it comparable to the previous record: 300 $\mu\text{m}/\text{min}$? See reference, Yuping Shi et al Na-assisted fast growth of large single-crystal MoS_2 on sapphire, Nanotechnology 2019 30 034002. I would like to see a more clear discussion on "the fast growth".

Ans: For our SCVLS reaction, Figure S11 b and c are the optical images of 0 min and 1 min growth. For 0 min growth, we only see the seeds of $\text{Na}_2\text{Mo}_2\text{O}_7$ with partly sulfurization (the sulfur concentration in the liquid is not enough to drive the reaction and precipitate the MoS_2 layer). For 1 min growth, the average edge length of the MoS_2 triangle is 370 μm . The average growth rate is 370 or 214 $\mu\text{m}/\text{min}$ (calculated by the edge length or the length from geometry center to tip), which is better than most reported references, and is better to the one reviewer provided (300 or 173 $\mu\text{m}/\text{min}$).

The reason why we did not compare the growth rate with other reports is that the calculation methods of growth rate are not very precise or consistent in many articles because they start counting the growth time as the temperature reach their target temperature. However, the growth may have already started during the ramping period (sulfur- and Mo-precursor would have sufficient vapor pressure before reaching the designed growth temperature), resulting in an overestimated growth rate, especially for methods using high temperature.

Figure S11. Growth rate of SCVLS. (a) Two different growth periods (0 and 1 minute) are used here. The as-grown optical images are shown in (b) and (c). Both scale bars are 200 μm .

2. “Conventional chemical vapor deposition methods for two-dimensional materials generally use an extremely small amount of precursor to render large single-crystal flakes, which inevitably causes low coverage of the materials on the substrate.” The judgment is only concluded from one reference, I didn’t think it can be generalized to “conventional chemical vapor deposition method”.

Ans: We have two references (ref.22 and 24) in Figure 3(g) to indicate this phenomenon (this trend is clear in many references but there are few reports which provided the chart of coverage vs amount of precursors in their manuscripts). There are some special cases such as substrate-induced single orientation growth (epitaxial) of 2D crystals. In this case, many small single crystals with the same orientation can merge into large crystals, so they can still use a higher amount of precursors to get the large-crystal film. Wafer-scale or large-grain epitaxial growth has been achieved in graphene and hBN (*Science* 2014, 344, 286-289 and *Nature* **2020**, 579, 219–223), but hasn't been achieved in TMD (arXiv preprint arXiv:1909.03502, 2019). We think the sentences we wrote in the abstract are general for most case in TMD CVD system, but we will replace the word "Conventional" into "most", "inevitably" into "usually" and "2D materials" into "transition metal dichalcogenides" to make it clear. We thank reviewer's reminder.

3. "An intermediate liquid phase Na₂Mo₂O₇ is formed through a eutectic reaction,". Please revise this sentence to "An intermediate liquid phase Na₂Mo₂O₇ is formed through a eutectic reaction of MoO₃ and NaF,".

Ans: Thanks for reviewer's suggestion. We have modified this.

4. "The field-effect transistors fabricated from the full-coverage films show high mobility (33 and 49 cm²V⁻¹s⁻¹ for the mono and bilayer regions)". It should be the average mobility and the value 33 cm²/Vs is conflict to the data present in Page 11: "90% of devices have pure monolayer channel and show high mobilities (34 ± 7 cm² V⁻¹s⁻¹) with a small variation of V_{th} (4.9±2.3 V)".

Ans: In the previous revised manuscript, we have fabricated hundreds of "new devices across the continuous film in Figure 6" to prove the advantage of the large-grain, continuous film. The devices are different from the original figure 5, so it is reasonable to have different mobility values from the data in figure 5.

5. Page 4: "However, because of the relatively low melting temperature of NaCl and rapid eutectic reaction, this method can only generate MoS₂ nanoribbons, which considerably limits its application." This comment totally misunderstands the paper. The reason for growing 1D MoS₂ nanoribbon only because of the low wettability between the growth substrate and the liquid droplets, the liquids prefer to keep spheres during the sulfurization process. In the reference 31,

wafer-scale 2D monolayer MoS₂ films can be grown with these liquid precursors, e.g., Na₂MoO₄.

Ans: Thanks for reviewer's reminder. We have modified this part in our manuscript.

Material synthesis and growth mechanism.

One obvious drawback in this part is that the authors use too many hypotheses and schematic illustrations instead of direct experimental data. That makes all the discussions weak and no roots.

6. Page 4: "Figure 1a and S1 show that a smooth MoO₃ layer was grown on c-plane sapphire through plasma-enhanced atomic layer deposition." There is no data indicating a smooth MoO₃ layer, only a schematic illustration.

Ans: We have added the AFM image of the MoO₃ grown by atomic layer deposition in Figure S1b. The roughness of the ALD MoO₃ is 0.25 nm, which is much better than the evaporated one shown in Figure S21a.

7. Page: "Figure S4 shows a full-coverage monolayer MoS₂ film grown on a sapphire of 3 × 3 cm² (size was only limited by the CVD tube size)". It is not a fully-coverage monolayer MoS₂ film. It also contains voids, thick layers. Please mention the ratio of monolayer, thick-layers, and voids.

Ans: The ratio of monolayer: thicker layers: voids are 92.3:7:0.7. Note that the voids are all in the edge region (<200 μm from the edge. We showed this edge image (Figure S4f, Figure S5f in this revision) with voids to let reader verify the contrast difference between region with and without MoS₂). It is full-coverage in the region far from edges.

Controlling the coverage and thickness of MoS₂.

8. The understanding of the growth mechanism can be wrong. We need more direct evidence, just propose a schematic illustration is meaningless.

Here are some core questions must be answered with direct evidence. The authors mentioned the later introduction of the sulfur, the Na₂MoO₇ will grow up to large droplet and lead to the growth of bilayer MoS₂. So please do more controlled experiments to prove it.

In control experiments without introducing sulfur, what are the area-density and size of the Na₂MoO₇ particles with different annealing times at the growth temperature?

Ans: In our previous revision (Figure S6, Figure S2 in this revision), we have used “experimental data” to show the MoS₂ was grown on water-soluble metal salt including NaF and Na₂Mo₂O₇. In addition, the porous SiO₂ also confirms our proposed mechanism.

Here we provided the data without sulfurization. Figure S15a is the structure of sample and the temperature ramping curve. The Na₂Mo₂O₇ particles started to appear at 750 °C (about 0.2-1 μm size, as shown in Figure S15b) and the size of droplets became larger as the temperature increased to 800 °C (about 1-10 μm size, as shown in Figure S15c), which indicated the liquid gradually rose up on the NaF surface. When we maintained at 800 °C for additional 2 minutes, the particle became larger and started wetting the surface (the wetting area is about 100-400 μm² as shown in Figure S15d), explaining why we can grow bilayer structure by delaying the timing of sulfurization. The droplet density is difficult to be extracted from the OM image because we cannot differentiate the contrast of liquid that is in or on the NaF matrix easily. However, it is a clear trend that the liquid piles up into large droplets and the density of droplets decreases for the high-temperature situations. The droplet density is much higher than the MoS₂ flake density after growth, which implies a self-capping mechanism as we have proposed in the manuscript.

Figure S15. Dynamic control of eutectic reaction (without sulfurization). (a)

Different heating time and temperature are used to control the size of droplets. (b), (c) and (d) are the as-grown optical images correspond to the blue, red and purple ramping curves shown in (a)

9. Page 7: “The average grain size of the full-coverage film was approximately 450 μm ”
The authors estimate the nucleation only happened in the first 1 minute, but the truth is the nucleation of MoS₂ can happen at the whole growth process. And the grain size of approximately 450 μm is strongly against the data presents in Figure S4.

Ans: In Figure S10, we showed the growth speed of the SCVLS method is about 370 $\mu\text{m}/\text{min}$. In Figure 3e or S10c, 82% of surface is covered by MoS₂ within 1-minute growth. From this result we can infer that the remaining 18% of the total area will be covered by MoS₂ film within the next minute, which makes nucleation of new MoS₂ flakes unlikely because it needs incubation time for MoS₂ nucleation, and we also don't see any Na₂Mo₂O₇ particles that have already come up on the NaF surface. Therefore, we estimate the grain size of the full-coverage film by linearly scaling the size of 1-minute growth. It is difficult to estimate the grain size from optical image of a fully-covered film because of invisible grain boundaries in the continuous film.

The image in Figure S4f (Figure S5f in this revision) is the region near the edge (we showed this image to verify the different contrast of region with and without MoS₂). The edge region usually has some non-uniform or incidental defect during process that causes defects and voids after growth. Nevertheless, the film is large-grain and full-coverage in the region far from edge.

10. Page 8: “Although there are some small islands on the film (Figure 3d)” I don't think thick layers with a dimension of hundreds of micrometers is “small”. From Figure 3d, we can see the thick layers are very large with a high ratio in the as-grown film. This non-uniformity will cause problems when integrating it into electronic devices.

Ans: Although we have some non-uniform regions with the thick area in the continuous film, the bottom of film consists of large-grain monolayer across the whole area still shows good electrical performance as demonstrated in Figure 6. The main novelty of this work is developing a new method that can grow continuous, large-grain and high-quality 2D thin film for applications. This thick area may be due to the oversaturated sulfur vapor and imprecise introducing time, which could be solved by using H₂S_(g).

11. Page 8: “the bottom continuous film can be peeled off and transferred to various substrates with the recently developed layer-resolved 2D material splitting

technique³⁴.” If the authors think the method reported in Ref. 34 is applicable to their samples, please demonstrated it with direct efforts. This kind of proposal is meaningless.

Ans: The main idea of this article is proposing a brand-new SCVLS growth method to grow high-quality, large-grain (record-high) and continuous film to solve the small-grain issue of continuous films with growth methods developed previously. Though we believe it is reasonable and stimulating to propose ideas to solve some problems in the manuscript for all readers, we have decided to remove this sentence since the demonstration of this method is beyond the scope of this paper.

Technical issue:

12. The mention of supplementary figures in manuscript is in quite random order. No mention of Figure S12-18 in the manuscript.

Ans: We have changed the order of supplementary figures in the manuscript and made an effort to mention every one of them in the manuscript.

Supplementary Information

13. Figure S1c: It should be “temperature ramping profile”.

Ans: We have changed it.

14. Figure S6: Please change “3-7 nm MoS₂ precursor” to “3-7 nm MoO₃”

Ans: We have changed it, thanks for the reminder.

15. Figure S7: What are the thickness of SiO₂ and MoO₃ layers?

16. Figure S8: What are the thickness of SiO₂, MoO₃ and NaF layers?

17. Figure S9: What are the thickness of NaF and MoO₃ layers?

Ans for 15-17: We have included all the relevant parameters in the corresponding figure captions.

Rebuttal Letter

18. Question 1: In Figure R1, the authors proposed the growth of tungsten-based TMDs and TMD alloys, but we DO want to see more experimental data on these aspects.

Ans: Because we do not have the tungsten source in our ALD system, we provided the data of MoSe₂ here to prove the SCVLS can be applied to different 2D materials. Figure S6a and b are the structure of precursor and the temperature ramping profile of the growth process. With 6 and 8 nm-thick and MoO₃ precursors, we got both large crystals and continuous film as shown in Figures S6c and d. The sharp photoluminescence (Figure S6e) of as-grown MoSe₂ confirms the monolayer property. The strong Raman peak (figure S6f) located at 241.6 cm⁻¹ indicates the A_{1g} peak of

crystalline MoSe₂. The as-grown MoSe₂ can also be transferred by dissolving the bottom metal salt by water as we did for SCVLS MoS₂. Figure S6g is the as-transferred MoSe₂ on silicon substrate. This data ensures the capability of SCVLS of growing different 2D materials.

Figure S6. SCVLS growth of MoSe₂. (a) Structure of precursor. (b) Temperature ramping profile of the process. (c) and (d) are the optical images of as-grown MoSe₂ with 6-nm and 8-nm MoO₃ precursors. Both scale bars are 150 μm. (e) Photoluminescence and (f) Raman spectra of as-grown MoSe₂. (g) Transferred MoSe₂ on the silicon substrate with a 300 nm thermal oxide layer. The scale bars is 150 μm

19. Question 2: The AFM results (Figure S12) only show small MoS₂ flakes, but cracks can be seen in large-grain MoS₂ films as shown in Figure S4.

Ans: The black lines in Figure S4 (Figure S5 in this revision) are not cracks of MoS₂. The direct proof of this is shown in Figure R1. The black line is continuous from the large MoS₂ single crystal to region that is not covered by MoS₂, which proves that this is not a crack of MoS₂. We think these lines are formed from the reaction between the materials with the sapphire since the lines are very directional and the only crystallized material in the system other than MoS₂. In addition, these cracks are not found after MoS₂ crystals are transferred.

Figure R1. (a)-(c) Optical images of as-grown MoS₂. Black lines that show up after growth are circled in red. The black line crosses continuously from MoS₂ to region that is not covered by MoS₂, indicating that the black lines are not cracks of MoS₂. Scale bars are 500 μm.

20. Question 11 and 17 references

1. Batch production of 6-inch uniform monolayer molybdenum disulfide catalyzed by sodium in glass, **Nature Communications 9(2018) 919**. This is very important progress in CVD growth of large-area (6-inch) MoS₂ film with a grain size of 400 μm in 8-min growth on a cheap glass substrate. This paper shows many advantages over the current paper. It must be included in Table 1 and Figure 3g. Intentionally overlook this important literature is unacceptable.

2. Kinetic modulation of graphene growth by fluorine through spatially confined decomposition of metal fluorides. Nature Chemistry 11(2019) 730-736. This is the first research paper that uses fluoride as growth promoters. It works well not only for the growth of TMDs but also for graphene and h-BNs. It is an important progress in the “salt-assisted growth of 2D materials”. It is also strongly related to this work, using NaF as a growth promoter.

3. Nanoscale 11(2019) 16122-16129. The citation in Reference should change from “Li, S. et al. Wafer-Scale and Deterministic Patterned Growth of Monolayer MoS₂ via Vapor-Liquid-Solid Method. arXiv Prepr. arXiv1906.05436 (2019).” to “Li, S. et al. Wafer-Scale and Deterministic Patterned Growth of Monolayer MoS₂ via Vapor-Liquid-Solid Method. Nanoscale 11 16122-16129

Ans: We did not intentionally ignore the reference reviewer provided. Based on the reviewer’s comment, we finally figured out the first paper reviewer 1 mentioned is Nature Communications 9(2018) **979** (Batch production of 6-inch uniform monolayer molybdenum disulfide catalyzed by sodium in glass), **instead of Nature Communications 9(2018) 919 he gave us in the first-round suggestion.** That was a huge misunderstanding and we definitely did not intentionally ignore this. As for the

comparison between these two work, the transport properties of the MoS₂ is inferior than what we presented (See table 1) and their grain size is also smaller. For second reference, because it is more related to graphene/hBN and the metal salt they used (BaF₂) is different from ours (NaF). Although they also used this method to grow WS₂/hBN, but they mentioned in their article that the mechanism for WS₂/hBN is not clear. That's why we did not include this reference in the first round. However, we would include it in the revised manuscript because it is the first one to use metal fluoride as reviewer mentioned. For the third reference, we have corrected the citation form of this reference.

Reviewer #2 (Remarks to the Author):

After reading the response letter thoroughly, as well as the revised manuscript, the reviewer feels this manuscript still falls short to the standard in nature communications for the following reasons.

1, The VLS method has been used previously.

Ans: We proposed the brand new SCVLS with a totally different mechanism. Also, we have mentioned about conventional VLS in our first-round manuscript.

2, The self-capping effect is not well controlled, as the multilayer region cannot be completely suppressed.

Ans: The reviewer has asked how to eliminate the flake under the current technique in his/her first-round question (question 3). We provided an idea of using H₂S_(g) to well control the sulfurization in the revised manuscript.

3, The monolayer quality has only incremental improvement over previous results in terms of mobility or current.

Ans: Our mobility and current are all comparable to the best reported CVD papers as shown in Table 1. The most important achievement is that we get the record-high grain size in continuous film with the new method.

4, The wafer scale result is not convincing with very few surface and material characterization. Previous work (ACS Nano 2017) has demonstrated even better growth results.

Ans: For the wafer-scale result, we have provided new OM images (Figure S4, Figure S5 in this revision) and electrical properties of the large-area continuous film (Figure 6) in our first-round revision. As for the reference reviewer provided, it does not include full citation, so we could not compare with it here.

5, The overall figure quality and data presentation is of low-quality, far below the standard of nature communications. The reviewer has raised this issue in the first round process and the quality has not been improved.

Ans: We have improved the quality of the whole Figure 5 in the first-round revision. For the second round, we improved the quality of Figures 1, 2 and 4, and also made all figures more readable by changing the form in Figure 2 and 3 in the revised manuscript.

Overall, this work lacks the novelty, significance and quality to be accepted in Nature Communications.

Ans: We have developed a brand-new method to grow high-quality, large-grain (record-high) continuous 2D film for application. It solves the small-grain issue for conventional CVD method. We believe our work has enough significance, novelty and quality.

In addition, here is the first-round comment from reviewer 2 him/herself. *“The authors report a self-capping method to fabricate large grain continuous MoS₂ mono- and bi-layers. **The fast growth rate and coverage of the resulting film are quite impressive. This work is interesting in general to the field and the approach adopted quite novel.** However, there are still some issues to be addressed.”* The criticism of novelty this round is weak based on reviewer’s words in the first-round comment.

Reviewer #3 (Remarks to the Author):

The questions have been addressed by the point to point responses of the author and I agree that this work is published in Nature Communications.

Ans: We thanks for reviewer’s previous comment and suggestion for this work.

REVIEWER COMMENTS

Reviewer #1 (Remarks to the Author):

All the comments are well addressed, I would like to recommend publication in Nature Communications.

Reviewer #2 (Remarks to the Author):

The authors provided some useful data in this round of review. However, it still fall short of a high quality paper, especially for a material intended for electronic devices.

1, The authors stressed several times that the grain size is record high. However, it is only twice large compared to previous results. In the material growth area, this is typically considered as incremental.

2, The novelty from the growth method itself, it is not the first time using spatially confined decomposition of metal fluorides from reviewer 1's comment. BaF₂ and NaF essentially play the same role in the growth and thus the difference is minimal.

3, How to eliminate the multilayer region is a huge problem. The authors only provide a hypothesis of using H₂S(g) which is groundless without any supporting data.

4, The electronic device data in figure 5 is quite poor even compared to previous work, especially in Fig. 6c. This dispersion is far from uniform as the authors claimed.

5, As shown in table 1, the data from this work has only incremental advantage compared to other work. The current is from a device biased at V_{ds}=8V, which clearly is very high in the common device applications, and very likely exceeding those from other references.

Overall, this work shows little advances in terms of material growth method, the material quality, and the devices based on this material. As a result, the reviewer does not suggest its publication in Nature Communications.

Response to reviewer

Here we replied all questions proposed by reviewer 2 as below.

1, The authors stressed several times that the grain size is record high. However, it is only twice large compared to previous results. In the material growth area, this is typically considered as incremental.

Ans: We think that the doubling grain size is a significant improvement especially for the large-grain and record-high condition (ex: 450 um v.s. 225 um is more prominent than the case for 4.5 um and 2.25 um).

2, The novelty from the growth method itself, it is not the first time using spatially confined decomposition of metal fluorides from reviewer 1's comment. BaF₂ and NaF essentially play the same role in the growth and thus the difference is minimal.

Ans: The functions of BaF₂ and NaF are very different because the eutectic reaction (formation of liquid phase **Na₂Mo₂O₇**) only occurs between MoO₃ and **alkali metal salt** instead of **alkaline earth metal salt**. Therefore, in that paper they did not form the liquid phase precursor. The mechanism is vastly different from our self-capping vapor-liquid-solid reaction (SCVLS). SCVLS involving self-capping and fast liquid transport effect for fast 2D materials growth is demonstrated for the first time in this manuscript.

3, How to eliminate the multilayer region is a huge problem. The authors only provide a hypothesis of using H₂S(g) which is groundless without any supporting data.

Ans: We have mentioned that the timing of sulfurization has a great effect on the layer number in Figure 4. The liquid forms a larger droplet if we introduce sulfur vapor late, resulting bi- or trilayer MoS₂ because the as-formed seed is buried inside the large droplet. The formation of large droplet is also proven in our previous revision (Figure S15). Both results support our SCVLS mechanism and point out the importance of the sulfurization timing. On the other hand, if the sulfur vapor is introduced too early, the vapor will diffuse into NaF and form non-fully sulfurized products in the NaF matrix, which blocks the liquid to go up. Figure S24(b) is the optical image of the growth product when introducing sulfur too early. The low sulfur concentration in the matrix results in the non-fully sulfurized products. Therefore, it is really important to introduce enough sulfur vapor as the liquid have just come up on the NaF surface to get a uniform monolayer. Because the sulfur vapor is generated from heating the solid sulfur powder, the amount of vapor and the introducing

timing is difficult to control (especially for sulfur, it starts evaporating before 100 °C, where is the region that is difficult to be well-controlled by the high-temperature furnace). Therefore, it is reasonable to propose the H₂S gas as the sulfur source, with which the timing of sulfurization can be well-controlled. We did not do this because H₂S is extremely toxic and is not allowed in our institute, but there are many articles that have used H₂S as the sulfur source to grow MoS₂ (ex: *2D Materials*, **2015**, 2.4: 044005 and *Journal of Physics: Condensed Matter*, **2016**, 28, 184002). In addition, we have also demonstrated uniform and monolayer MoSe₂ films with other vapor source (selenium, starts evaporating at 210 °C, a temperature that we can controller a lot better with our high temperature furnace) in our previous revision (Figure S6d), proving the SCVLS can grow uniform continuous film if we can control the reaction vapor well.

Figure S24. (a) Temperature ramping profile of the growth with a very early sulfurization (b) Optical image of the as-grown product with a very early sulfurization condition.

Figure S6. SCVLS growth of MoSe₂. (a) Structure of precursor. (b) Temperature ramping profile of the process. (c) and (d) are the optical images of as-grown MoSe₂ with 6-nm and 8-nm MoO₃ precursors. The scale bars are 150 μm. (e) Photoluminescence and (f) Raman spectra of as-grown MoSe₂. (g) Transferred MoSe₂ on the silicon substrate with a 300 nm thermal oxide layer. The scale bar is 150 μm.

4, The electronic device data in figure 5 is quite poor even compared to previous work, especially in Fig. 6c. This dispersion is far from uniform as the authors claimed.

Ans: Our device performance is good compared with most studies as shown in Table 1 (especially when considering all properties at the same time). For the uniformity issue, we didn't claim high uniformity for the continuous film in the manuscript. We use one hundred FETs across the 1.5 X 1.5 cm continuous film to characterize the uniformity as-shown in figure 6. We did mention the non-uniformity is resulted from some multilayer regions, but the overall electrical performance is still good because of the large-grain, continuous bottom layer (some multilayer regions that causes non-uniformity even have better electrical performance), which is important for real applications. We only claimed good uniformity in the large single crystals with TEM (Figure S4) and FET (Figure S20) measurement, which is important to prove the single crystal property of the non-perfectly-triangular flake grown by SCVLS.

Table 1. Comparison of the grain size, mobility, maximum current density, and on/off ratio of monolayer MoS₂ in previous research and this study. All transport data were obtained from the back-gate FET for comparison.

Growth method	Maximum Grain size (single crystal)	Mobility (cm ² V ⁻¹ S ⁻¹)	Maximum current density (μA μm ⁻¹)	On/off ratio	Reference
CVD MoS ₂ (MoO ₃ +S, substrate control) ⁴⁹	200 μm	25		10 ⁷	ACS Nano 2015 , 9, 4611
CVD MoS ₂ (MoO ₃ +S, flow control) ⁵⁰	300 μm	30		10 ⁶	Adv. Sci. 2016 , 3, 1500033.
CVD MoS ₂ (MoCl ₅ +DMS, NaCl catalyst) ⁵¹	50 μm	10.4		10 ⁷	Nanotechnology 2017 , 28, 465103
CVD MoS ₂ (MoO ₃ +S, molten Na:glass) ³²	563 μm	24	123	10 ⁹	Appl. Phys. Lett. 2018 , 113, 202103
CVD MoS ₂ (MoO ₃ +S with PTAS salt) ⁴²	200 μm	35	270	10 ⁷	Nano Lett. 2018 , 18, 4516
CVD MoS ₂ (Mo foil+S, soda-lime glass) ⁵²	400 μm	11.4		10 ⁶	Nat. Commun. 2018 , 9, 979
SCVLS MoS ₂	1.1 mm	33	230	5×10 ⁸	This work

5, As shown in table 1, the data from this work has only incremental advantage compared to other work. The current is from a device biased at V_{ds}=8V, which clearly is very high in the common device applications, and very likely exceeding those from other references.

Ans: We have already discussed this in our first-round revision. The electric field is about 5.7 V/μm in our device, which is very common to reach in the nowadays short channel devices. We apply high voltage to extract the maximum current density, which is an important parameter to demonstrate the material and device quality and this saturation region is the operation region for the modern digital circuit. (ex: *ACS Nano* **2015**, 9, 8, 7904-7912. They use 8V in a 0.8 μm (10 V/μm) and *Nano letters*,

2018, 18, 4516-4522.) The high maximum current density can only be achieved with both high-quality materials and good contact properties. If one of them is bad, the current will saturate or even degrade very soon (the channel is likely to break because of the current heating issue if the contact resistance is large under a high-field condition), so it is unsuitable to say it is only an incremental advantage here. In our device, $V_{ds} = 8V$ is in the saturation region (Figure 5e), which is also the region every literature in the table reported in, so we think the table stands as a fair comparison. Many previous studies did not discuss this detail because of they have contact or material issues. We provided this measurement to further confirm the quality of our device and material. In addition, the on-off ratio is still $\approx 10^8$ even we only applied 2 V in this device and the mobility would not increase when a larger bias is applied in.

REVIEWERS' COMMENTS:

Reviewer #2 (Remarks to the Author):

The authors addressed the first three of the comments. However, the last two comments were not answered correctly. The figure quality and style in Fig.5 and 6c are far below the standard microelectronic device characteristics. Specifically, Fig. 5b doesn't reveal any information of the device structure with poor image contrast. Fig.5c and 5d shows channel conductance without normalization while 5e and 5f are normalized current. The correct notation for gate voltage is V_{gs} for gate-to-source voltage instead of V_G . The source to drain voltage should be denoted as V_{DS} instead of V_{SD} . The V_{gs} is 40V, 45V, 50V and 50 V in Fig 5c to 5f, respectively. And even in Fig 5f along, the x-axis for black and red curves are not measured within the same range of V_G . And typically, the channel length should not be L , but L_{ch} or L_g . In the caption of Fig. 5f, this should be called transfer characteristics in semi-log plot, not Log-plot of the gate-dependent current density. In Fig.6b, the optical image provides little information of the device. In Fig.6c, the transfer characteristics are considered pretty poor quality for any such devices. In a word, the reviewer feels such electrical data of the devices would not get anywhere close to publication in any device related journals, let alone prestigious journals like NCOMM. The authors should at least be consistent with device measurement, make the correct plot, name the right notations and they can find any of those information in classic semiconductor textbooks.

Response to reviewer 2's question

Reviewer #2 (Remarks to the Author):

The authors addressed the first three of the comments. However, the last two comments were not answered correctly. The figure quality and style in Fig.5 and 6c are far below the standard microelectronic device characteristics. Specifically, Fig. 5b doesn't reveal any information of the device structure with poor image contrast. Fig.5c and 5d shows channel conductance without normalization while 5e and 5f are normalized current. The correct notation for gate voltage is V_{gs} for gate-to-source voltage instead of V_G . The source to drain voltage should be denoted as V_{DS} instead of V_{SD} . The V_{gs} is 40V, 45V, 50V and 50 V in Fig 5c to 5f, respectively. And even in Fig 5f along, the x-axis for black and red curves are not measured within the same range of V_G . And typically, the channel length should not be L , but L_{ch} or L_g . In the caption of Fig. 5f, this should be called transfer characteristics in semi-log plot, not Log-plot of the gate-dependent current density. In Fig.6b, the optical image provides little information of the device. In Fig.6c, the transfer characteristics are considered pretty poor quality for any such devices. In a word, the reviewer feels such electrical data of the devices would not get anywhere close to publication in any device related journals, let alone prestigious journals like NCOMM. The authors should at least be consistent with device measurement, make the correct plot, name the right notations and they can find any of those information in classic semiconductor textbooks.

Ans:

1. Reviewer did not point out what is wrong with our previous answers for the last two questions (we answered them very carefully last time).
2. Reviewer says the device performance in fig 6c is poor but it is actually quite similar to data in fig. 5c (mobility: 33 vs 37 $\text{cm}^2\text{V}^{-1}\text{S}^{-1}$) which we have shown in the comparison table that the performance is top of its class (benefitted from bottom large-grain and continuous bottom layer). This is a wrong accusation without any support references. The deviation of performance comes from both the non-uniformity of material (Mostly the outlier with higher current/performance because of multilayer clusters. We have also discussed the effect of cluster on the device performance in figure S21) and we have also claimed a promising route to further improve the uniformity by using the $\text{H}_2\text{S}_{(g)}$ precursor in our

manuscript and the reviewer 1 and 2 are both satisfied with this point in the previous revision.

3. We have changed the notation of L , V_g , V_d into L_{ch} , V_{GS} and V_{DS} as reviewer required. The optical contrast of Figure 5b is improved and the fine feature of Figure 6b is shown in Figure S18 (we added this annotation in the figure caption).
4. Reviewer misunderstood that our measurement in fig. 5c-d are not normalized. In fact, it is normalized and has an actual unit of S/sq (siemens square) but is normally simplified as siemens in most papers for devices.
5. Reviewer mentioned the different scan range of gate voltage in figures. However, it is typical in a device paper to have different scan ranges of gate voltage for devices with different threshold voltage to show the full picture of how the device operates (Generally, we would not use a large V_{GS} for all measurement in order to avoid the dielectric breakdown). Ex: For figure 5d, a larger sweep range (+45V) was used to clearly show the metal-insulator transition. For figure 5e, a larger gate voltage (+50V) is used to show the highest current density. For figure 5f, a larger negative voltage (-50V) was used to clearly show the off state of the bilayer sample.